# Interaction between aerosol and thermodynamic stability within the PBL during the wintertime over the North China Plain: Aircraft observation and WRF-Chem simulation

Hao Luo[1], Li Dong[2], Yichen Chen[3,4*], Yuefeng Zhao[5*], Delong Zhao[3,4], Mengyu Huang[3,4], Deping Ding[3,4], Jiayuan Liao[1], Tian Ma[1], Maohai Hu[2*], Yong Han[1,6*]

[1]Advanced Science & Technology of Atmospheric Physics Group (ASAG), School of Atmospheric Sciences, Sun Yat-sen University & Southern marine Science and Engineering Guangdong Laboratory, Zhuhai 519082, China
[2]School of Electronic and Optical Engineering, Nanjing University of Science and Technology, Nanjing 210094, China
[3]Beijing Weather Modification Office, Beijing 100089, China
[4]Beijing Key Laboratory of Cloud, Precipitation and Atmospheric Water Resources (LCPW), Beijing Meteorological Bureau, Beijing 100089, China
[5]School of Physics and Electronics, Shandong Normal University, Jinan 250014, China
[6]Key Laboratory of Tropical Atmosphere-Ocean System (Sun Yat-sen University), Ministry of Education, Zhuhai 519082, China

* *Correspondence to*: Yichen Chen (chenyichen@bj.cma.gov.cn); Yuefeng Zhao (yuefengzhao@sdnu.edu.cn); Maohai Hu (hmh@njust.edu.cn) & Yong Han (hany66@mail.sysu.edu.cn)

**Abstract.** Aerosol-planetary boundary layer (PBL) interaction has been proposed as a key mechanism for stabilizing the atmosphere and exacerbating surface air pollution. Although the understanding of this process has progressed enormously, its magnitude and impact remain uncertain and vary widely concerning aerosol types, vertical distributions, synoptic conditions, etc. In this study, our primary interest is to distinguish the aerosol-PBL interaction of absorbing and scattering aerosols under contrasting synoptic patterns and aerosol vertical distributions. Detailed in-situ aircraft (KingAir-350) measurements and online coupled model Weather Research and Forecasting with Chemistry (WRF-Chem) simulations are explored over the North China Plain (NCP). Furthermore, a long-term PBL stability trend from 1980 to 2020 over the NCP is also investigated. The aircraft measurements and surface observations show that the surface air pollution over the Baoding City on 3 January is heavier than that on 4 January, 2020. In addition, the aerosols are restricted to the low layer on 3 January, whereas the aerosols mix more homogeneous upwards on 4 January. Thereupon, we focus on the two days with distinct synoptic circumstances, PBL stability, and aerosol vertical distributions over the NCP. According to the WRF-Chem modeling, the synoptic pattern over the Baoding City differs between the two days. The prevailing wind direction is opposite with a southwest wind on 3 January and a northeast wind on 4 January. The results indicate that the synoptic condition may affect the PBL thermal structure, thus affecting the aerosol vertical distribution. Additionally, the sensitive numerical experiments reveal that the light-absorbing and light-scattering aerosols have different effects on altering the PBL thermal structure. The inhibition effect of scattering aerosols on the PBL appears to be independent of the aerosol height distribution and solely depends on its concentration. However, aerosol-PBL feedback of absorbing aerosols is highly dependent on its

vertical distribution. Besides the two-day case investigation, the analysis of the modeling results for nearly one month from January 3 to 30, 2020 in Baoding city yields a more robust and representative conclusion. Our analysis highlights that we should principally concentrate on controlling the emissions of scattering aerosols under the stable stratification while cooperating to control the emissions of scattering and absorbing aerosols in an unstable stratification. Moreover, the long-term inter-annual variation in PBL stability shows a strong correlation with the East Asian Winter Monsoon, which seems to be valuable in determining which pollutants to target in different monsoon years and attaining more precise air pollution control. Based on the numerical simulations and observational constraints, a concept scheme description has been concluded to deepen our recognition of the interactions between thermodynamic stability and aerosols within the PBL over the NCP region.

## 1 Introduction

Ambient air pollution has been one of the major environmental issues in China, particularly in the highly populated and industrialized areas, such as the North China Plain (NCP) (Chan and Yao, 2008; Sun et al., 2014; Zhang, Q., Zheng, Y. et al., 2019a; Fan et al., 2020; Luo et al., 2021). Severe and persistent air pollution episodes commonly break out in the presence of high aerosol emissions and unfavorable synoptic circumstances (e.g. low wind speed, high moisture, and stable stratification), posing great threats to human health (Che et al., 2019; Zhang, X. et al., 2019b). The interaction of aerosols with the planetary boundary layer (PBL), which is regarded to be a critical process for stabilizing the atmosphere and worsening surface air pollution, has been widely explored in the context of aerosol weather and climate effects (Li, Z. et al., 2017a; Su et al., 2020; Hung et al., 2021).

Aerosols contribute to considerable uncertainties in interpreting and quantifying the Earth's radiative budget and hydrological cycles through aerosol-radiation interactions (ARI) and aerosol-cloud interactions (ACI) (Rosenfeld et al., 2014; Luo et al., 2019; Zhao et al., 2019; Letu et al., 2021). Despite the great progress in observational and numerical studies of ARI and ACI over the recent decades, correctly quantifying the aerosol radiative effect (ARE) on the weather and climate systems remains a challenge. The principal reason for this challenge is the inadequate understanding of strong variations in aerosol types, loadings, and vertical distributions, as well as the complex mechanisms among large-scale synoptic patterns, local-scale planetary boundary layer (PBL) structures and AREs (Wang, H. et al., 2015; Li, Z. et al., 2017a; Huang et al., 2018; Su et al., 2020).

The thermodynamic stability of the PBL dictates the planetary boundary layer height (PBLH), thereby dominating the vertical dissipation of surface pollutants to some extent (Zhang, Q., Ma, X. et al., 2009; Zhang, W. et al., 2018; Su et al., 2020). Aerosols, in turn, have vital feedback on the stability of the PBL, depending on their properties, particularly those of light-absorbing aerosols (e.g., black, brown and organic carbon) (Huang et al., 2018; Menon et al., 2002; Wang, Z. et al., 2021). The feedback mechanisms between aerosol and PBL are critical in modulating air pollution outbreaks (Ding et al., 2013; Ding et al., 2016; Huang et al., 2018; Ma et al., 2020). This feedback process can be divided into two aspects, relating

to surface cooling and atmospheric heating through ARE. On the one hand, the suspended aerosols in the atmosphere attenuate the incident solar radiation, leading to an overall cooling of the surface, diminishing surface sensible heat flux, and impeding the PBL development, which is known as the umbrella effect (Ma et al., 2020; Shen and Zhao, 2020). On the other hand, light-absorbing aerosols can trap solar energy and heat the atmosphere, which has different effects depending on the aerosol vertical distribution. For one thing, the absorbing aerosols in the low layer of the atmosphere strongly heat the near-surface layer to form a well-mixed PBL, which is recognized as the stove effect (Ma et al., 2020). For another, the absorbing aerosols in the upper layer of the atmosphere strengthen the PBL inversion intensity and weaken its development, which is noticed as the dome effect (Ding et al., 2016).

In addition, the large-scale synoptic patterns disturb the sensitivity of the aerosol-PBL feedback mechanism. PBLH is not only influenced by the local-scale surface properties (e.g., surface cover types, sensible heat fluxes, latent heat fluxes, etc.), but also regulated by large-scale synoptic forcing (Hu et al., 2016; Miao et al., 2020). Synoptic conditions can alter the PBL thermodynamic stability through cold or warm advection, thus affecting the aerosol vertical dispersion (Zhang, J. et al., 2012; Ye et al., 2016; Miao et al., 2019). Furthermore, due to the prevailing south winds, aerosols emitted in the southern regions can be carried to NCP via cross-region transport, and the increased aerosols eventually modulate the PBL thermodynamic stability (Du et al., 2020; Ma et al., 2020). On the inter-annual scale, previous studies reported that the wintertime air quality is closely connected with the East Asian Winter Monsoon (EAWM), and the strengthening (weakening) of EAWM is typically associated with changes in representative circulation patterns, which can improve (worsen) regional air quality (Niu et al., 2010; Zhang, Y. et al., 2016).

According to the aforementioned efforts, most of them primarily concentrated on the aerosol-PBL interactions neglecting the simultaneous influences of the synoptic condition, the aerosol type and vertical distribution. With numerical simulations and observational constraints, the roles of synoptic pattern, PBLH, aerosol type and vertical distribution in aerosol-PBL interactions warrant further investigation. To address this issue, this study attempts to analyze the effects of synoptic forcing and aerosol types on the PBL thermodynamic stability, aerosol vertical distribution, and aerosol-PBL interaction by using in-situ aircraft measurements, surface observations, and on-line coupled model Weather Research and Forecasting with Chemistry (WRF-Chem) simulations. On the 3[rd] and 4[th] of January, 2020, two days with distinct synoptic conditions, PBL stability, and aerosol vertical distributions over NCP are analyzed. Five parallel numerical experiments are conducted to investigate the AREs of scattering and absorbing aerosols under different aerosol vertical distributions. Furthermore, to compensate for the two-day case investigation, an analysis of the modeling results for nearly one month in Baoding city from January 3 to 30, 2020 gives a more robust and representative conclusion. Moreover, the long-term inter-annual variation in PBL stability from 1980 to 2020 over the NCP region is examined, and its driving factors are figured out. The conclusions appear to be beneficial in determining which pollutants to target in different weather conditions and achieving more precise controls of air pollution.

The remainder of this paper is organized as follows. Section 2 describes the data and methods used in this study. Section 3 presents and discusses the aerosol-PBL interactions, as well as the factors of synoptic forcing and aerosol types. The conclusions and summary are provided in section 4.

## 2 Data and Methods

### 2.1 Flight experiments and observational data

#### 2.1.1 Intensity aircraft observation experiments

To study the vertical distributions of aerosol concentrations and meteorological conditions, the intensity aircraft observation experiments (aircraft model: KingAir-350, which is displayed in Fig. 1e) were carried out to measure the vertical profiles on January 3 and 4, 2020. The terrain height and three-dimensional flight routes are shown in Fig. 1 during the study periods. The flight routes on the two days were almost identical, which is conducive to the comparative analysis. The KingAir-350 took off and climbed to about 3000 m over the Shahe airport (40.1° N,116.3° E), which is located in Beijing City. Afterward, the aircraft proceeded towards the southwest for about 150 km, arriving in Baoding City and performing vertical measurements ranging from around 650 m to 3000 m.

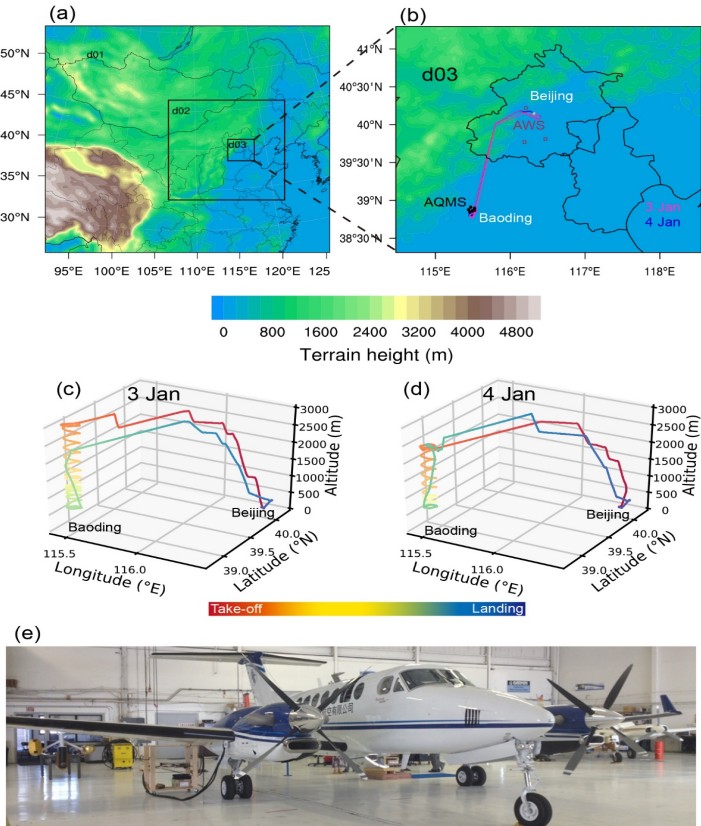

**Figure 1: WRF-Chem domains with the terrain height and flight routes. (a) The outermost domain (d01) and the two embedded domains (d02 and d03); (b) the innermost domain (d03); (c-d) the three-dimensional flight routes on January 3 and 4, 2020, respectively; (e) the KingAir-350 aircraft. The black dots and brown boxes in panel (b) show the locations of the air quality monitoring sites (AQMS) and automatic weather stations (AWS), respectively. The red and blue lines in panel (b) represent the horizontal flight routes on January 3 and 4, 2020, respectively.**

In this study, the aerosol number concentrations and size distributions were measured by a passive cavity aerosol spectrometer probe (PCASP) mounted on the wingtip of the KingAir-350 aircraft. PCASP detected aerosol size distributions ranging from 0.1 to 3.0 μm at 1 Hz. Given the substantial noise of the first channel, we considered all the spectral bands except the first one, i.e. 0.11 to 3.0 μm, in this study (Twohy et al., 2005). A seven-wavelength (370, 470, 520, 590, 660, 880, and 950 nm) aethalometer (AE-33) was used to detect aerosol absorption. Because black carbon (BC) is the predominant absorber at 880 nm (Ran et al., 2016), measurements at this wavelength were taken to represent BC concentration. In addition, an integrated meteorological measurement system (AIMMS) was employed to measure the aircraft location and ambient meteorological conditions (e.g. temperature, wind, etc.). The detailed information of aircraft observation is summarized in Table 1.

**Table 1: Information of the KingAir-350 aircraft flight times and instruments.**

| Flight time | Date | Take-off time | Landing time |
|---|---|---|---|
| | January 3, 2020 | 1600 LST | 1830 LST |
| | January 4, 2020 | 1530 LST | 1820 LST |
| | Instrument | Objects | Manufacturer |
| Observation instrument | PASCP-100X | Aerosol number concentration, size distribution, etc. | Droplet Measurement Technologies |
| | AE-33 | BC concentration | Magee Scientific |
| | AIMMS-20 | Location, temperature, wind speed, etc. | Advantech Research Inc. |

### 2.1.2 Ground-based observations

Surface meteorological observations at three ground-based automatic weather stations (AWS) in Beijing City were collected for model meteorological validation, which includes the hourly datasets of surface pressure, 2 m temperature, 10 m wind speed, etc. In addition, air quality observations, including hourly ground-based $PM_{2.5}$ concentration measurements at six air quality monitoring sites (AQMS) in Baoding City, were used in this study to validate the model performance. The observational air pollutant data was collected from the China National Environmental Monitoring Center

 Ground-based observations on January 3 and 4, 2020 were applied in this study, and the locations of AWS and AQMS are shown in Fig. 1b.

## 2.2 Reanalysis data

The data of isobaric temperature taken from the European Centre for Medium-Range Weather Forecasts (ECMWF) ERA5 reanalysis dataset was used to summarize the long-term thermal structure of PBL in North China Plain. The temperature data of seven isobaric surfaces between 1000 hPa and 850 hPa were considered to be within the PBL. The height between 1000 hPa and 850 hPa is approximately equivalent to the height below 1.5 km. The sea level pressure from the ERA5 reanalysis product was used to calculate and describe the intensity of the EAWM and Siberian High (SH), which are demonstrated to be the aspects that vigorously govern the PBL thermal structure over the NCP in this study. Here, the monthly average data during the wintertime (December, January and February) from 1980 to 2020 with a grid of 0.25°×0.25° was adopted.

## 2.3 WRF-Chem model

In this study, we simulated the aerosol concentrations and meteorological conditions by using the WRF-Chem model version 3.9, which is an online-coupled three-dimensional Eulerian chemical transport model considering complex physical and chemical processes (Grell et al., 2005). The three nesting domains with horizontal resolutions of 27, 9 and 3 km, respectively, are shown in Fig. 1a. There were 29 vertical layers extended from the ground to the top pressure of 50 hPa in the model, with more than 15 layers located below 3 km to fully describe the vertical structure of the PBL. The simulations were conducted from 1200UTC on December 25, 2019 to 0000UTC on January 31, 2020, with the first 8.5 days as spin-up time for chemistry. The model was run with an 84 hours model cycle, with the first 12 hours discarded as spin-up time and the last 72-hour results used for the final analysis. The chemical outputs from previous runs were used as the initial conditions for the subsequent overlapping 84-hour simulation. The simulations of the case study were carried out from 0000 UTC on January 2 to 1800 UTC on January 4, 2020 with the first 16 hours as the model spin-up time, and the chemical outputs from the previous run were used as the initial conditions.

The 6-hour National Centers for Environmental Prediction (NCEP) global final (FNL) reanalysis fields with a spatial resolution of 1°×1° were input for the model initial and lateral boundary meteorological conditions. Anthropogenic emissions were adopted from the Multi-resolution Emission Inventory for China (MEIC) in 2017 developed by Tsinghua University (http://meicmodel.org), whose emission sources were classified into five sectors: industrial process, power plants, residential combustion, on-road mobile sources, and agricultural activities (Li, M., Liu, H. et al., 2017b; Zheng et al., 2018). The Model of Emissions of Gases and Aerosols from Nature (MEGAN) was used online in the simulation to calculate the biogenic emissions (Guenther et al., 2006).

The model physical configurations we used in the simulation included the Morrison double-moment microphysics scheme (Morrison et al., 2005), the RRTMG radiation scheme (Iacono et al., 2008), the Yonsei University (YSU) boundary

layer scheme (Noh et al., 2003) and the Noah land surface scheme (Ek et al., 2003). PBLH in the YSU scheme is determined from the Richardson bulk number method. Model for Simulating Aerosol Interactions and Chemistry (MOSAIC) was applied as the sectional aerosol scheme, with aerosols specified as 8 size sections (bins) ranging from 39 nm to 10 μm (Zaveri et al., 2008). The extinction, single-scattering albedo, and asymmetry factor of aerosols were computed as a function of wavelength and three-dimensional positions. Each chemical constituent of the aerosol was linked to complex indices of refraction. The refractive indices of the aerosols were calculated using volume averaging for each size bin, and the Mie theory was used to derive the extinction efficiency, the scattering efficiency, and the intermediate asymmetry factor. Aerosol optical properties were then determined by summarizing all size bins (Fast et al. 2006). The refractive indices of various aerosol components were reported in Barnard et al. (2010).

To investigate the interactions between aerosol and PBL thermodynamic stability over the NCP, five parallel numerical experiments were conducted as presented in Table 2. The lapse rate below 1.5 km was used to quantify the PBL thermodynamic stability. In the control experiment (denoted as the EXP_Ctrl), the anthropogenic emissions were performed at the normal level, and the aerosol optical properties were calculated at each time step and then coupled with the short- and long-wave radiative transfer model. In the EXP_WoF, the emissions were the same as the EXP_Ctrl, except that ARE was not included. In the other three experiments, ARE was taken into consideration, but the emissions varied. Specifically, the BC emission was subtracted in the EXP_WFexBC, the BC emission was 20 times higher in the EXP_WF20BC, and the total aerosol emissions were strengthened to 20 times in the EXP_WF20Aer. To eliminate the influence of cloud condensation nuclei under various emission scenarios, the aerosol indirect effect was turned off in the modeling.

Table 2: Descriptions of five parallel WRF-Chem numerical experiments.

| Experiment | Description |
|---|---|
| EXP_Ctrl | Control experiment, with the aerosol radiative feedback. |
| EXP_WoF | Without any aerosol radiative feedback. |
| EXP_WFexBC | With the aerosol feedback excluding the contribution of BC. |
| EXP_WF20BC | With the aerosol feedback including 20 times the BC emission. |
| EXP_WF20Aer | With the aerosol feedback including 20 times the total aerosol emission. |

**2.4 Statistical methods for model validation**

Correlation coefficient ($R$), mean bias (MB), normalized mean bias (NMB), mean error (ME) and root mean square error (RMSE) were applied to assess the WRF-Chem model veracity in simulating meteorological parameters and air pollutants against the ground-based observations with the following equations (Itahashi et al., 2015; Granella et al., 2021):

$$R = \frac{\sum_{i=1}^{n}(S_i - \bar{S})(O_i - \bar{O})}{\sqrt{\sum_{i=1}^{n}(S_i - \bar{S})^2 \sum_{i=1}^{n}(O_i - \bar{O})^2}}, \tag{1}$$

$$MB = \frac{1}{n}\sum_{i=1}^{n}(S_i - O_i), \tag{2}$$

$$NMB = \frac{\sum_{i=1}^{n}(S_i - O_i)}{\sum_{i=1}^{n}(O_i)}, \tag{3}$$

$$ME = \frac{1}{n}\sum_{i=1}^{n}|S_i - O_i|, \tag{4}$$

$$RMSE = \sqrt{\frac{1}{n}\sum_{i=1}^{n}(S_i - O_i)^2}, \tag{5}$$

where $S_i$ and $O_i$ are the simulated and observed parameters, respectively, $n$ is the total number of the values used for validation, and $\bar{S}$ and $\bar{O}$ are the averages of the simulation and observation, respectively.

## 3 Results and Discussion

### 3.1 Validations of meteorological parameters and aerosol concentrations

Considering that the weather conditions play important roles in pollutant transport, aerosol dispersion, and chemical reactions in the atmosphere, it is crucial to validate the simulated meteorological parameters with the observations. Fig. 2 shows the validation of meteorological parameters between the modeled and observed results, and model evaluations suggest that the WRF-Chem model is able to simulate the weather characteristics in the NCP region. The simulated temporal series and magnitudes of temperature, pressure and wind speed generally agree well with the ground-based and vertical aircraft observations. The accuracy of simulation during the daytime is superior to that at night, which is instrumental in the intention of discussing the PBL thermodynamic stability and ARE in the daytime. In addition, validations of aerosol concentrations between the modeling and in-situ observations are shown in Fig. 3. Both simulation and observation display a high level of air pollution on 3 January, and good air quality on 4 January 4. The vertical profiles of BC concentration suggest a good simulation performance, which can characterize the vertical variations and daily differences (Fig. 3a). The modeled surface $PM_{2.5}$ mass concentrations in Baoding City compare well with the ground-based measurements, especially during the daytime (Fig. 3b).

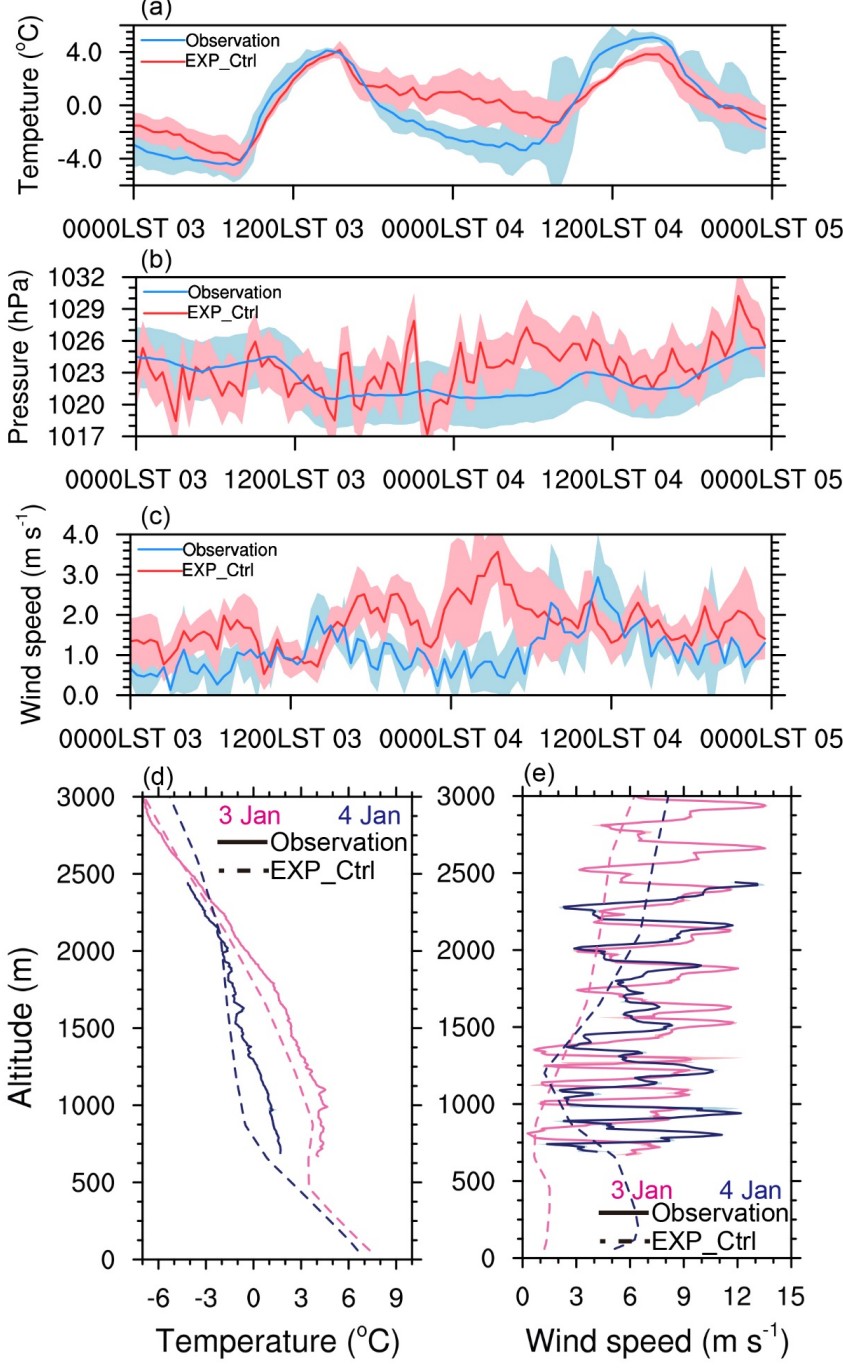

Figure 2: Validation of meteorological parameters. (a-c) Time series of 2m temperature, surface pressure, and 10m wind speed in Beijing City between the modeling (EXP_Ctrl) and ground-based observation, respectively; (d-e) vertical profiles of temperature and wind speed in Baoding City between the modeling (EXP_Ctrl) and aircraft observation, respectively. The shaded areas indicate the error bars (standard deviation).

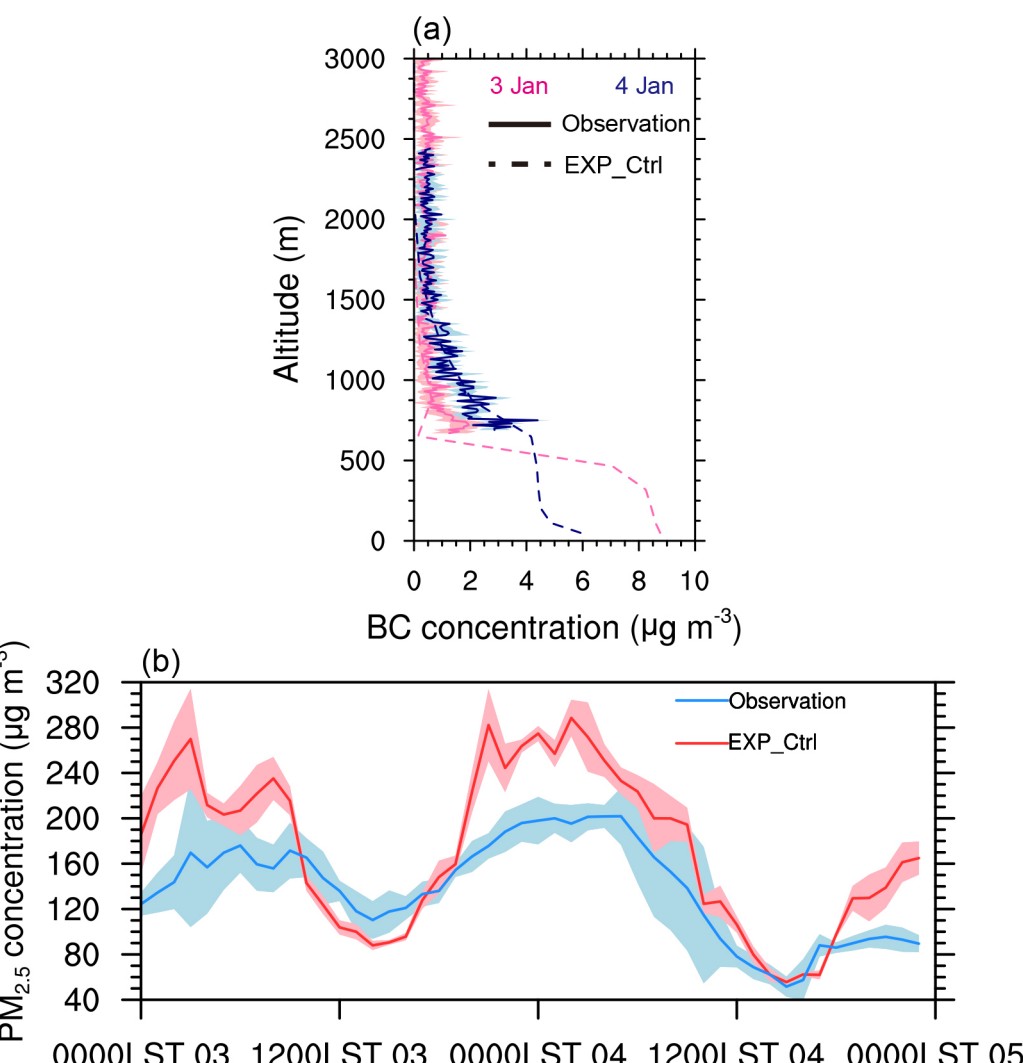

**Figure 3: Validation of aerosol concentration between the modeling (EXP_Ctrl) and in-situ observations. (a) aircraft measured BC concentration vertical distributions; (b) ground-based observed PM2.5 concentration. The shaded areas indicate the error bars (standard deviation).**

The statistical evaluations of model performance between in-situ observations and EXP_Ctrl are concluded in Table 3. Underestimation of temperature exists in both the surface level and vertical profile, though the *R* between the ground-based (aircraft) observation and EXP_Ctrl is 0.87 (0.98) that indicates the model well describes the temperature variation. The surface pressure is well simulated with an MB of 0.81 and a small NMB of 0.08%. Slight overestimation of 10 m wind speed (0.40 m s$^{-1}$) in the simulation might be due to the unresolved topography in the WRF model, which is also demonstrated in the previous studies (Jiménez et al., 2013; Li, M., Wang, T. et al., 2021a). The wind speed profile suggests an underestimation, which could be attributable to flaws in aircraft wind speed measurement. The statistical validations of BC

concentration vertical profiles show an $R$ of 0.67 and a total MB of -0.18 µg m$^{-3}$. The statistical validations of PM$_{2.5}$ mass concentration indicate an $R$ of 0.79, a total MB of -4.91 µg m$^{-3}$, and an NMB of 4.69% during the daytime (Table 3). Therefore, in this study, we consider that the WRF-Chem simulation is in line with the observation and can capture the weather characteristics as well as the general distributions and variations in air pollutants.

**Table 3: Statistical evaluations of the WRF-Chem performance between in-situ observations and EXP_Ctrl.**

| Platform | Variable | Observation | EXP_Ctrl | $R$ [a] | MB [a] | NMB [a] | ME [a] | RMSE [a] |
|---|---|---|---|---|---|---|---|---|
| Ground-based [b] | Temperature (°C) | 3.25 | 2.37 | 0.87 | -0.87 | -26.90% | 0.95 | 1.11 |
| | Pressure (hPa) | 1021.95 | 1022.76 | 0.42 | 0.81 | 0.08% | 1.44 | 1.81 |
| | Wind speed (m s$^{-1}$) | 1.14 | 1.54 | 0.29 | 0.40 | 6.97% | 0.49 | 0.64 |
| | PM$_{2.5}$ (µg m$^{-3}$) | 104.77 | 99.86 | 0.79 | -4.91 | 4.69% | 17.64 | 20.31 |
| Aircraft [c] | Temperature (°C) | -0.60 | -1.33 | 0.98 | -0.73 | 12.17% | 0.76 | 0.98 |
| | Wind speed (m s$^{-1}$) | 6.77 | 4.23 | 0.40 | -2.54 | 37.52% | 3.41 | 4.15 |
| | BC (µg m$^{-3}$) | 0.69 | 0.51 | 0.67 | -0.18 | -26.09% | 0.39 | 0.47 |

[a] $R$: correlation coefficient; MB: mean bias; NMB: normalized mean bias; ME: mean error; RMSE: root mean square error.

[b] Ground-based observation is from 1000LST to 1800LST on January 3 and 4, 2020.

[c] Aircraft measurement is from 650 m to 3000 m on January 3 and 4, 2020.

## 3.2 Aerosol vertical characteristics and PBL thermal structures

Despite the large-scale weather condition, the local-scale PBL thermal structure is also reported as a critical aspect that substantially affects the surface aerosol loading (Zhang, Q., Ma, X. et al., 2009). The PBL thermodynamic stability dictates the PBLH, thereby dominating the vertical dissipation of surface pollutants to some extent. Therefore, the detailed synoptic condition, PBL thermal structure, and their influences on the aerosol vertical characteristics are discussed in the following subsections.

### 3.2.1 Vertical thermal structure and synoptic condition

The temporal evolution of the EXP_Ctrl simulated temperature profile is presented in Fig. 4, which reflects the variation in the thermal structure and stability during the study period. The results indicate that the thermal structures differ significantly between the two days. It is noticed that the temperature on 3 January is warmer than that on 4 January, suggesting different wintertime synoptic characteristics. Furthermore, the average lapse rate below 1.5 km is lower on 3 January, with lapse rate values of 3.62 °C km$^{-1}$ and 5.38 °C km$^{-1}$ during the aircraft observation periods on 3 and 4 January, respectively.

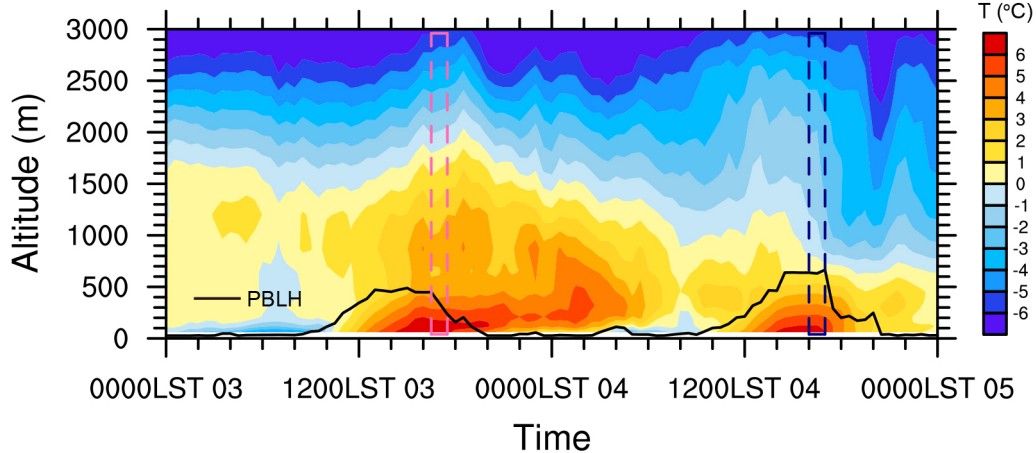

**Figure 4: Temporal evolution of the EXP_Ctrl simulated temperature profile over the Baoding City. The pink and blue dashed boxes correspond to the aircraft observation periods on 3 and 4 January, respectively. The black line indicates the planetary boundary layer height (PBLH).**

The thermal structure of the atmosphere is closely related to and highly sensitive to the synoptic condition over the NCP, especially during the wintertime (Yang et al., 2016; Miao et al., 2019). Figs. 5a and 5b display the surface synoptic patterns during the aircraft observation periods on 3 and 4 January, respectively. In the adjacent regions of the Baoding City, the wind fields are practically opposite in these two periods, and the contrast is also intuitively shown in Fig. 5c with the vertical profile difference in the meridional wind speed. The prevailing surface wind direction on 3 January is southwest, while on 4

January is northeast. The south wind at the low layer warms the atmosphere below 1.5 km on 3 January, whereas the north wind leads to a cooling effect on 4 January, thus there appears a large temperature difference manifested in Fig. 4. Furthermore, the south wind warms the atmosphere, particularly the aloft layer at about 1 km altitude, leading to a stable thermal stratification below 1 km and a low PBLH on 3 January.

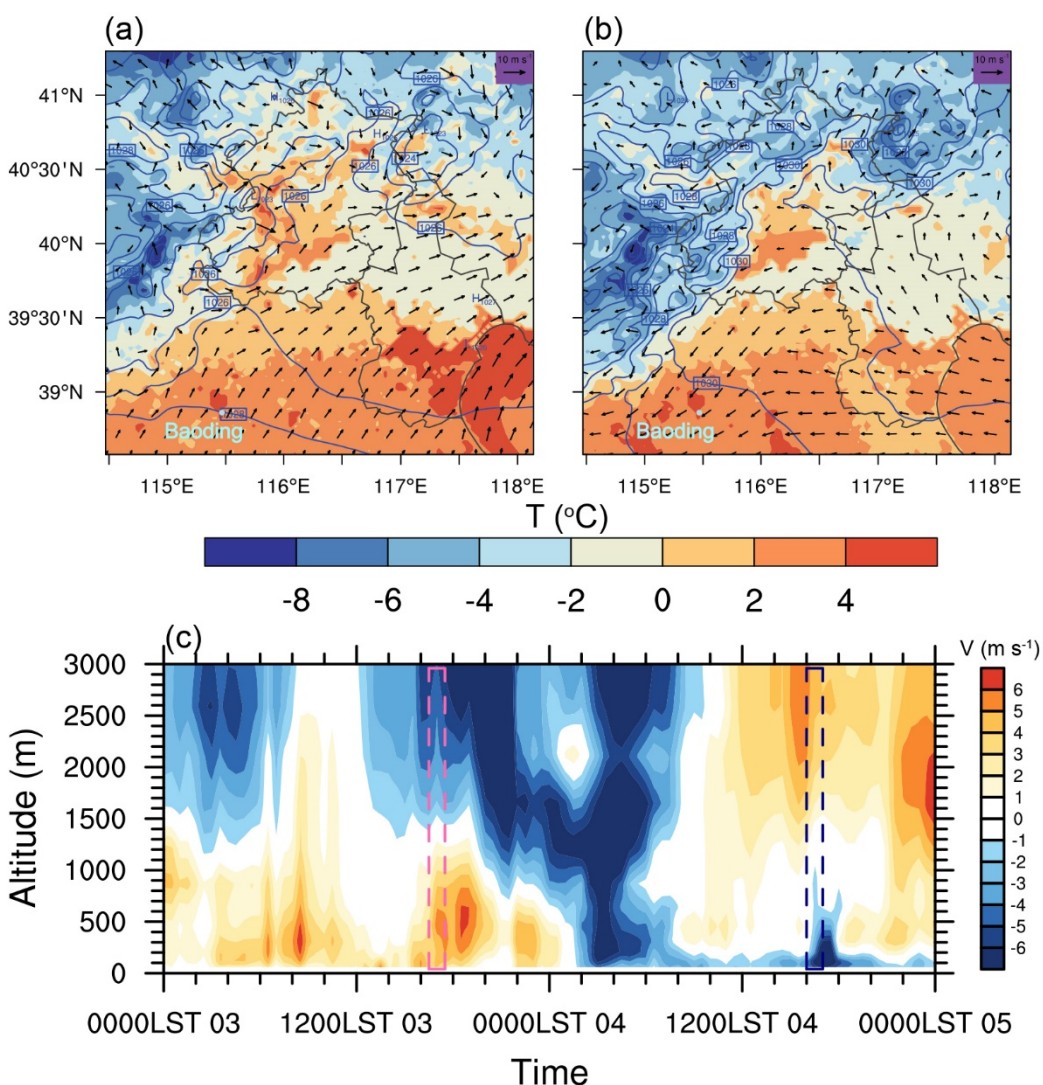

Figure 5: The EXP_Ctrl simulated synoptic patterns during the study period. (a-b) the surface wind, temperature, and pressure field during the aircraft observation periods on 3 and 4 January, respectively; (c) the temporal evolution of the meridional wind speed over the Baoding City. The green dots in (a-b) denote the location of the Baoding City. The pink and blue dashed boxes in (c) correspond to the aircraft observation periods on 3 and 4 January, respectively.

### 3.2.2 Observed and simulated aerosol vertical distributions

As demonstrated in Fig. 6, contrasting aerosol vertical distributions are observed with aircraft during the afternoon of January 3 and 4, 2020, and the specific times are shown in Table 1. The accumulation mode particles (0.1-1 μm) dominate the aerosol number concentrations during the study period, which is primarily emitted by anthropogenic pollutants, such as BC aerosol loadings (Wu, Y. et al., 2017). On 3 January, larger amounts of aerosol particles are constrained to the low layer of the atmosphere (below 650 m), while the pollutants decrease sharply above 650 m and gradually tend to a stable low value.

On 4 January, the aerosol number concentrations below 650 m are lower than that on 3 January, but higher when the height is above 1000 m. This contrast is attributed to the differences in atmospheric stability (Su et al., 2020). The aloft aerosol layer between 1.0 km and 1.7 km altitude on 4 January is related to the PBL vertical mixing transport. The great difference in lapse rate shown in Fig. 4 leads to the disparity in atmospheric stability and aerosol dispersion ability. On 4 January, the more unstable thermal stratification promotes the dispersion of the near-surface pollutants, and meanwhile, exacerbating the

pollution on the elevated layer.

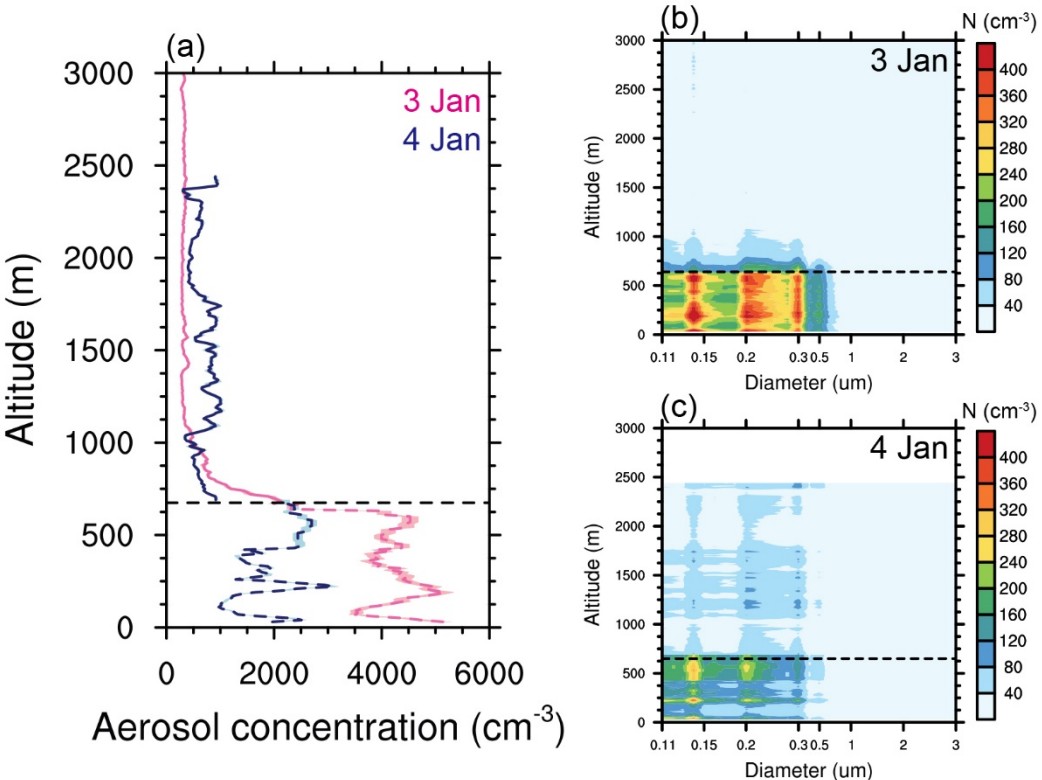

**Figure 6: Aircraft observations of aerosol vertical distributions in the afternoon during the flight. (a) the aerosol number concentrations on 3 and 4 January, the dashed and solid lines denote the observations over Beijing and Baoding, respectively; (b-c) the aerosol size distributions on 3 and 4 January, respectively. The shaded areas in (a) indicate the error bars (standard deviation).**

**The horizontal black dashed lines are the reference altitude of 650 m.**

     The temporal evolution of the EXP_Ctrl simulated vertical profiles of total aerosol number concentrations and BC mass concentrations are shown in Fig. 7. Both profiles have similar features, indicating that anthropogenic fine particles like BC, are the dominant pollutants, which is consistent with the particle size observations exhibited in Figs. 6b and 6c. The diurnal variation in the aerosol profile is strongly driven by the evolution of PBLH, which is well examined in the previous studies

(Qu et al., 2017; Huang et al., 2018). The pollutants are well constrained to the near-surface by the stable PBL at nighttime,

while the vertical distribution of aerosol particles is more homogenous by turbulent mixing in the afternoon. In addition, on 4 January, the vertical mixing of aerosols is more homogenous, transporting aerosols to a higher layer, which implies a more unstable thermal stratification and a higher PBLH.

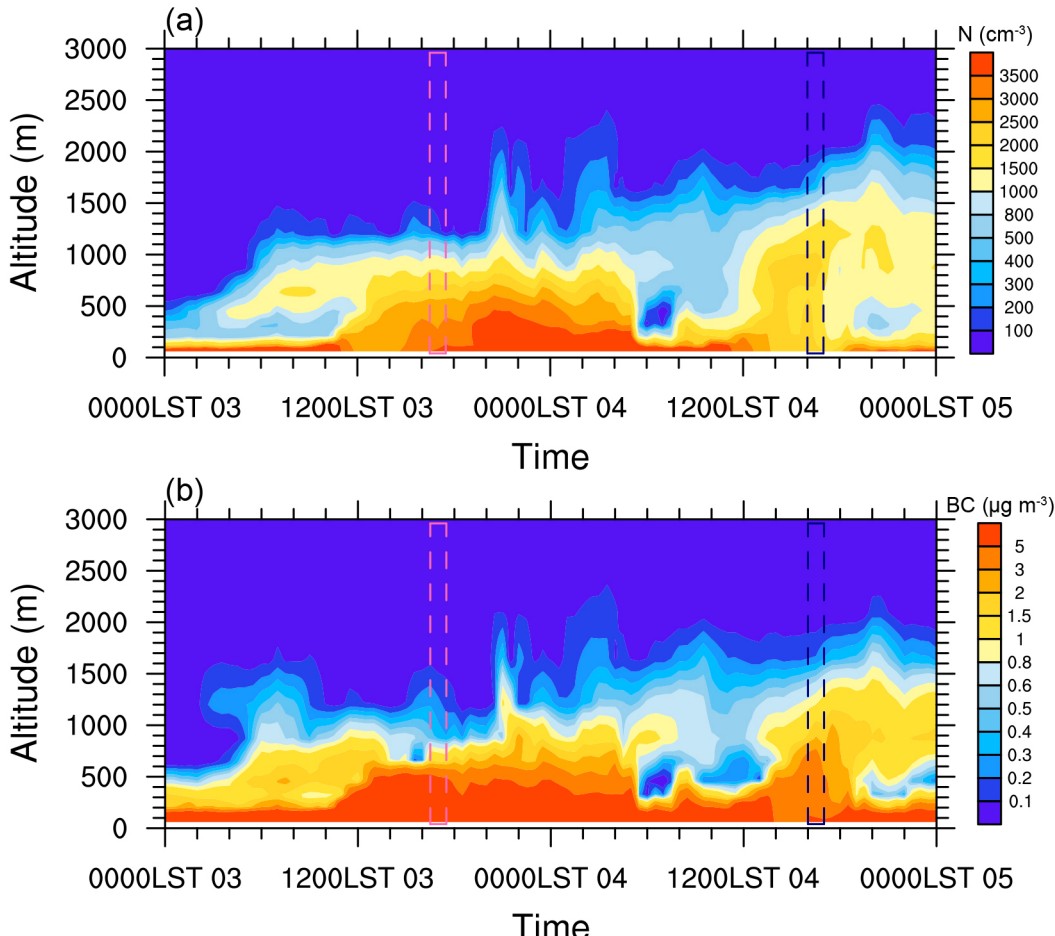

**Figure 7: Temporal evolution of the EXP_Ctrl simulated aerosol vertical distributions over the Baoding City. (a) the aerosol number concentration (particle diameter: 0.15-2.5 µm); (b) the BC concentration. The pink and blue dashed boxes correspond to the aircraft observation periods on 3 and 4 January, respectively.**

The contrasting aerosol vertical distributions between the two days are interpreted from two aspects. From the perspective of aerosol regional transport, the air quality is remarkably altered by the synoptic condition (Zhang, Y. et al., 2016; An et al., 2019). The accumulation of pollutants over NCP is contributed by regional transport from the polluted southwest region via the northward movement of the air mass, whereas the cessation of pollution over NCP is caused by prevailing northern air mass and then the pollution hotspot moves to the south (Luo et al., 2021). Therefore, the observed and simulated high surface pollution loading on 3 January is externally affected by the unfavorable synoptic condition, which is

consistent with the south wind demonstrated in Fig. 5. From the perspective of aerosol vertical distribution, the surface air
pollution loading is efficiently driven by the thermal structure and stability of the PBL (Miao et al., 2020; Li, Q. et al., 2021b). On 3 January, the prevailing surface south wind leads to a stable stratification, restraining aerosol vertical dispersion and finally exacerbating the surface air pollution. Besides, the north wind condition on 4 January has the opposite effect, which reduces the surface aerosol loading.

In conclusion, the stable stratification related by the prevailing surface south wind limits the PBL development and
exacerbates the surface air pollution by restraining the aerosols to the low layer. Conversely, the unstable stratification generated by surface north wind promotes the PBL turbulent mixing and lightens the surface aerosol concentration via vertical diffusion.

### 3.3 Aerosol and PBL thermodynamic stability interactions

### 3.3.1 ARE on PBL thermal structure

PBL structure modulates the aerosol vertical distribution by turbulent mixing, and inversely, the suspended aerosols may also modify the PBL thermal structure by ARE to some extent. Fig. 8 presents the temperature variation affected by ARE, and it is noticeable that light-absorbing aerosols heat the atmosphere while light-scattering aerosols contribute to a cooling effect. On 3 January, aerosols are constrained to the low layer during the daytime, and the overall ARE results in a warming effect below 1 km. The conspicuous heating suggests that the leading role of absorptive BC aerosols on the
variation in PBL thermal structure when coexisting with other types of scattering aerosols on 3 January. On 4 January, due to the strong turbulence mixing (as demonstrated from the temperature profile in Fig. 4), aerosols are carried to the aloft layer. The overall ARE results in a cooling effect below 0.6 km and a warming effect between 0.6 and 1.5 km. When the aerosols are transported to a high altitude, the absorbing and scattering particles inhibit the incident solar radiation from reaching the low layer and exhibit a cooling effect, while the absorbing aerosols heat the upper layer. The contrasting aerosol vertical
distribution caused by the varied PBL thermal structure between the two daytimes results in different AREs, therefore, it is quite crucial to examine the aerosol vertical structure when evaluating the ARE.

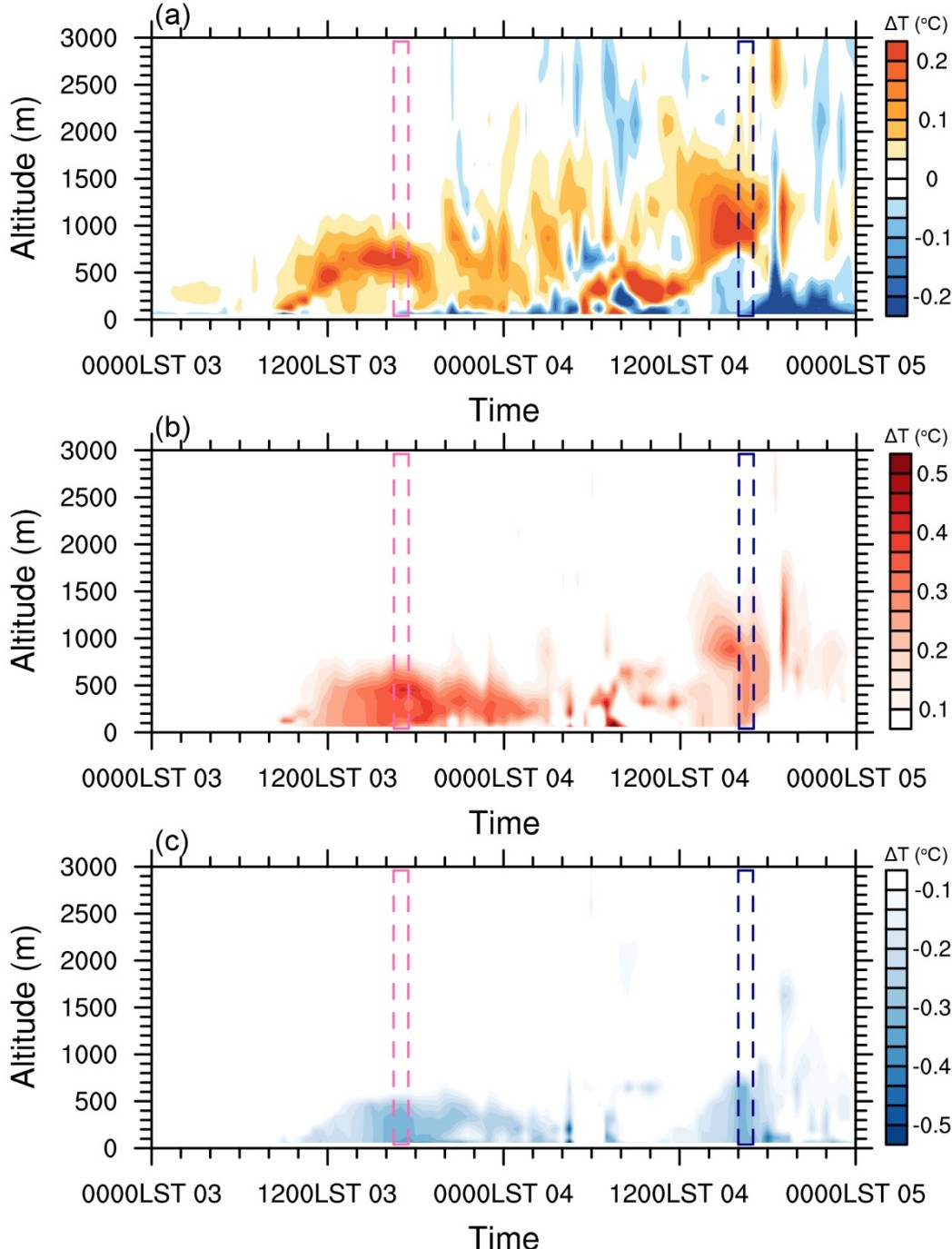

**Figure 8: Temporal evolution of temperature profile variation influenced by aerosol radiative effect (ARE). (a) ARE by all the aerosols (EXP_Ctrl – EXP_WoF); (b) ARE by BC (EXP_Ctrl – EXP_WFexBC); (c) ARE by other aerosols (EXP_WFexBC – EXP_WoF). The pink and blue dashed boxes correspond to the aircraft observation periods on 3 and 4 January, respectively.**

The simulated ARE modify the thermal structure with a temperature variation of less than 0.5 °C shown in Fig. 8, to amplify the signal of ARE when discussing its interactions with PBL, we further performed two experiments, one with strong overall aerosol loading (EXP_WF20Aer) and the other only with strong BC aerosol loading (EXP_WF20BC). The simulated thermal structures are presented in Fig. 9. The amplified simulations reveal a temperature variation of less than 2.5 °C, which is consistent with the altered magnitude of aerosol emissions. ARE by only absorptive BC aerosols heat the low layer more remarkable when compared with the role of overall aerosols, suggesting that the coexists of absorbing and scattering aerosols partially conceal the signal of ARE. This result is owing to the opposite optical roles of the two aerosol types in the low layer, as well as the strong inhibition of the incident solar radiation by the aerosols in the upper layer.

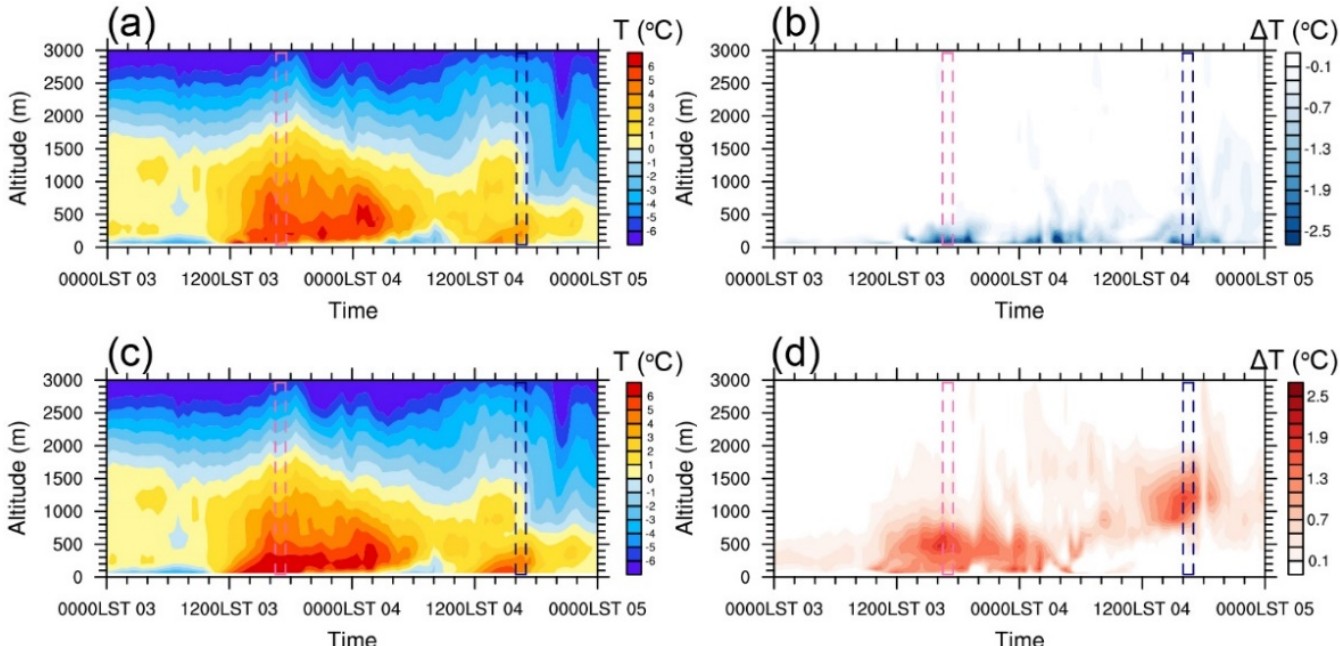

**Figure 9: Temporal evolution of temperature profile with (a) high aerosol emissions (EXP_WF20Aer) and (c) high BC emissions (EXP_WF20Aer); the temporal evolution of temperature variation profile influenced by (b) high scattering aerosol loadings (EXP_WF4Aer – EXP_WF20BC) and (d) high absorbing aerosol loadings (EXP_WF20BC – EXP_Ctrl). The pink and blue dashed boxes correspond to the aircraft observation periods on 3 and 4 January, respectively.**

### 3.3.2 ARE on the PBLH, lapse rate, and surface pollution

Since ARE can modulate the PBL thermal structure, the related impacts caused by the thermal variation are discussed here. Fig. 10 shows the time series of the simulated lapse rate, PBLH, and surface carbon monoxide (CO) concentration under five parallel experiments.

Generally, on 3 January, the PBLH is lower and the surface air pollution is heavier than those on 4 January. Influenced by the southwest winds, the warmer air mass can be guided to NCP (Fig. 5a), enhancing the thermal stability and restraining

the growth of the PBL (Fig. 10b). On the one hand, from the local-scale view, the lower PBLH constrains the air pollution to the near-surface; on the other hand, from the regional transport view, the southwest prevailing winds can transport the pollutants emitted from upstream plain regions to NCP, which further worsen the surface air quality (Fig. 10c). However, the relatively higher PBLH and the northeast winds on 4 January are able to reduce the surface air pollution and inversely carry the pollutants to the higher altitude with the strong turbulent mixing. The disparate synoptic conditions regulate the vertical aerosol structures and further affect the ARE and other related variables.

The distinct aerosol types contribute to varied AREs when different aerosol vertical distributions are dominant. The lapse rate and PBLH are closely associated with the vertical thermal structure. A warmer low layer and a colder upper layer induce a higher lapse rate and PBLH, while a colder low layer and a warmer upper layer decrease the lapse rate and PBLH. On 3 January, the pollutants are restricted to the low layer, but the absorbing and scattering aerosols play the opposite roles. Ma et al. (2020) has confirmed that the radiative heating of high concentration absorbing aerosols near the surface will offset and exceed the reduction of surface sensible heat flux, resulting in near-surface heating (stove effect). The low-layer heating, together with the surface buoyancy flux, invigorates the turbulent mixing and leads to a higher lapse rate and PBLH (Ma et al., 2020). On the contrary, the scattering aerosols cool the low layer and inhibit the growth of the PBL (umbrella effect), causing a lower lapse rate and PBLH (Figs. 10a and 10b). On 4 January, the aerosol vertical profile differs from that on 3 January, with a huge number of pollutants being carried to the high altitude by the unstable stratification. The absorbing and scattering aerosols both prevent the solar radiation to the low layer (umbrella effect), while the absorbing aerosols also heat the upper layer (dome effect), generating a lower lapse rate and PBLH (Figs. 10a and 10b).

The variations in lapse rate and PBLH further alter the surface air pollution by the thermodynamic and dynamic effect, and high PBLH ordinarily lightens surface air pollution. Due to the CO is a relatively stable pollutant that is hardly interactions with other pollutants and the main sources of CO are primary emissions from anthropogenic pollution and open biomass burning (Chi et al., 2013), we use CO concentration to evaluate the variation in surface air pollution caused by the ARE in this study. The surface CO concentration in the EXP_WF20Aer is divided by a factor of 20 to match other experiments since the emissions are magnified 20 times. The results presented in Fig. 10c indicate that the surface air pollution variation is closely related to the PBLH variation and driven by the aerosol vertical dispersion. When the stable stratification confines the absorbing aerosols to the low layer, the stove effect destroys the stability, enhancing the thermal convection and decreasing the surface air pollution. The absorbing aerosols in the upper layer or the scattering aerosols in both the low and upper layers stabilize the atmosphere, deteriorating the surface air quality.

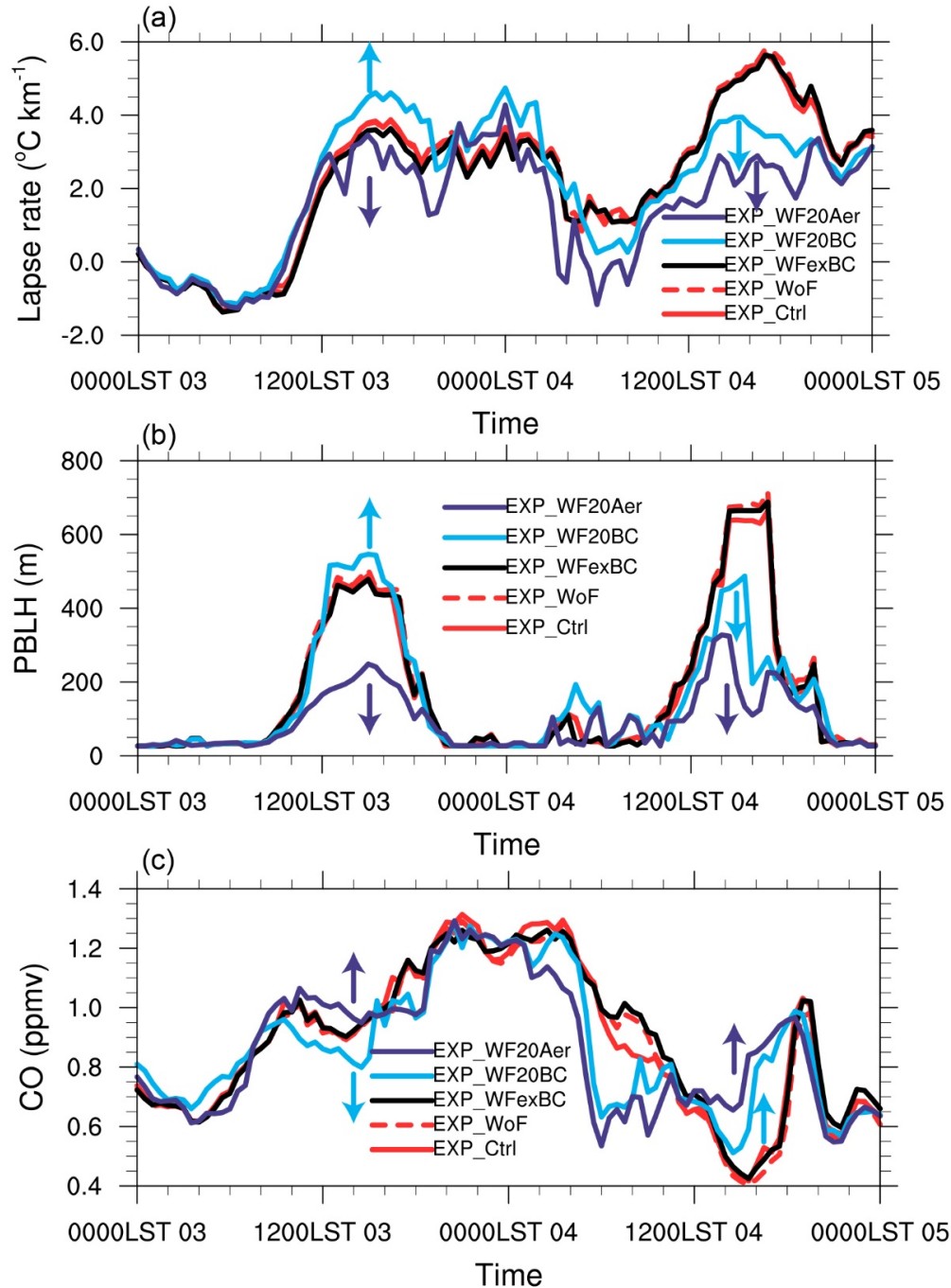

**Figure 10: Temporal evolution of the (a) lapse rate within 1.5 km, (b) planetary boundary layer height (PBLH), and (c) surface CO concentration simulated by the five parallel experiments. The upward or downward arrows indicate an increase or decrease compared to the EXP_Ctrl.**

375

### 3.3.3 Statistical properties of the PBL and AREs under different synoptic conditions

It is noticeable that different aerosol vertical distributions between the two days contribute to distinct AREs due to the synoptic condition and PBL thermal stability differences from the measurements and simulations. In particular, the absorptive BC aerosols have both stove and dome effects, which affect the PBL thermal structure. Here, we further analyze the modeling results for nearly one month from January 3 to 30, 2020 in Baoding city to give a more significant and representative conclusion.

Fig. 11 shows the correlations between the daily average 10 m meridional wind speed, lapse rate within 1.5 km, and PBLH. The negative correlation between the 10 m meridional wind speed and the lapse rate within 1.5 km (Fig. 11a) suggests that the increased south wind stabilizes the PBL, whereas the strong north wind destabilizes the PBL. The variation in lapse rate has a direct impact on the development of the PBL, as evidenced by the PBLH modification shown in Fig. 11b. Fig. 12 compares the distinct vertical distributions of aerosols caused by north and south winds. Samples with a daily average wind speed within $\pm$ 0.05 m s$^{-1}$ are discarded to avoid the north-south reverse of wind direction in a day. Eventually, 16 days with the prevailing north wind and 8 days with the prevailing south wind are used. The result indicates that the synoptic condition influences the PBL thermal structure, thereby affecting the vertical dispersion of aerosols. The warm and polluted air is carried to the NCP by the south winds, which stabilize the PBL, exacerbating the surface air pollution. The cold and clean air is carried to the NCP by the north winds, forming an unstable stratification and transporting pollutants to the upper layer. The previous study has also verified the conclusion that wind field and PBLH tend to be correlated, which means meteorologically favorable horizontal and vertical dispersion conditions are likely to occur together (Su et al., 2018).

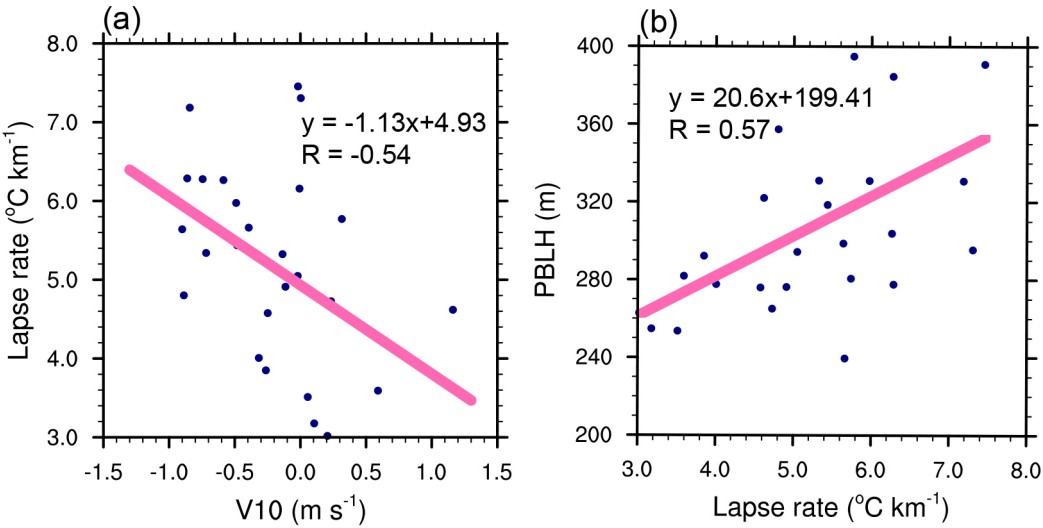

**Figure 11: Scatter plots of the correlations between (a) 10 m meridional wind speed (positive: south wind; negative: north wind) and lapse rate within 1.5 km and (b) lapse rate within 1.5 km and PBLH. The data are daily averages from January 3 to 30, 2020.**

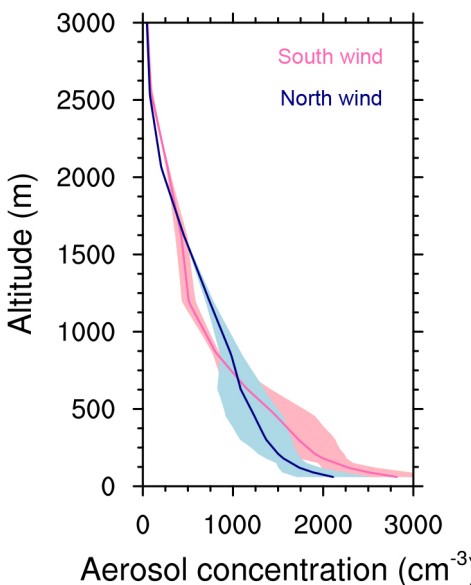

**Figure 12: Vertical distributions of the aerosol number concentrations (particle diameter: 0.15-2.5 µm) under the prevailing south wind and north wind, respectively. The shaded areas indicate the error bars (standard deviation).**

When evaluating the AREs of light-absorbing and light-scattering aerosols, the temperature profile variations show various patterns due to differences in aerosol concentration and vertical distribution caused by synoptic conditions, particularly the wind direction. Fig. 13 demonstrates that light-absorbing aerosols heat the atmosphere while light-scattering aerosols contribute to a cooling effect. Aerosols are constrained to the low layer under south wind conditions, and the BC aerosols result in a warming effect below 1 km (stove effect), while the scattering aerosols cool the layer below 0.6 km (umbrella effect). In contrast, the PBLs exhibit strong turbulence mixing when influenced by the north winds, and the aerosols are carried to the aloft layer. The aloft scattering particles prevent incident solar radiation from reaching the low layer, resulting in cooling effects below 1 km (umbrella effect), whereas the aloft absorbing aerosols heat the upper layer between 0.5 and 1.5 km (dome effect). The remarkable aerosol effects under south winds attribute to the accumulation of aerosols under adverse weather conditions. The contrasting aerosol vertical distributions caused by the varying synoptic conditions lead to different AREs, which is consistent with the results obtained on January 3 and 4, 2020.

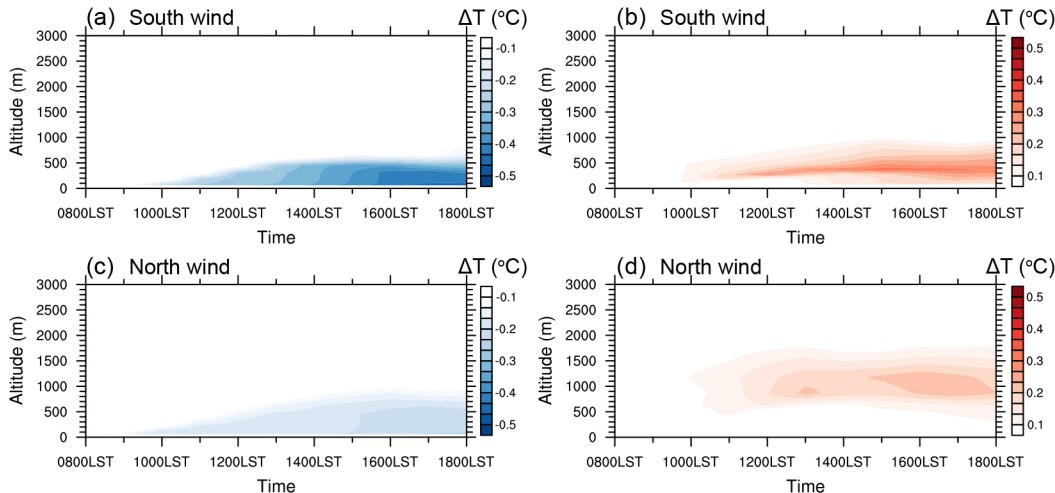

**Figure 13: Temporal evolution of the temperature profile variation influenced by aerosol radiative effect (ARE). (a) ARE by other aerosols (EXP_WFexBC – EXP_WoF) and (b) ARE by BC (EXP_Ctrl – EXP_WFexBC) under the prevailing south wind; (c) ARE by other aerosols and (d) ARE by BC under the prevailing North wind.**

Furthermore, based on the nearly one-month simulations, we quantify the variations in lapse rate within 1.5 km and PBLH under different synoptic conditions, respectively, caused by absorptive BC aerosols and other light-scattering aerosols. The results in Fig. 14 reveal that the BC stove effect induces a 0.04 °C km$^{-1}$ increase in lapse rate within 1.5 km and a 3 m increase in PBLH under the stable stratifications with the prevailing south winds. However, the BC dome effect causes a 0.085 °C km$^{-1}$ decrease in lapse rate within 1.5 km and a 3 m decrease in PBLH under the unstable stratifications with the prevailing north winds. The umbrella effect of scattering aerosols in both stable and unstable conditions reduces the lapse rate by about 0.15 °C km$^{-1}$ and reduces the PBLH by about 3.5–4 m. The vertical distribution of absorbing aerosols has a significant impact on their aerosol-PBL feedback. The absorbing aerosols concentrated in the low layer have a strong radiative heating effect on the atmosphere develop the PBL in the case of stable weather patterns under the influence of the south wind. The absorbing aerosols in the upper layer heat the atmosphere and inhibit the development of the PBL in the case of unstable weather patterns under the influence of the north wind. The inhibition effect of scattering aerosols on the PBL, on the other hand, is independent of the aerosol height distributions and is solely dependent on the aerosol concentrations.

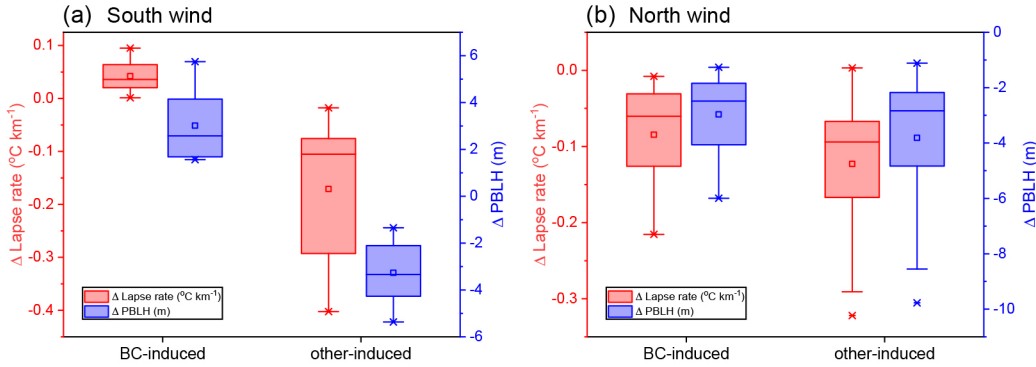

**Figure 14: Box plots of the variations in lapse rate within 1.5 km and PBLH influenced by the BC and other aerosols under (a) the prevailing south wind and (b) the prevailing north wind, respectively. The squares represent the mean values, the horizontal lines inside the boxes are the medians, and the bottom and top sides of the boxes represent the first and third quartiles. The whiskers are the minimum (maximum) values within 1.5 interquartile ranges of the lower (upper) quartile. The asterisks indicate the minimum (maximum) values.**

## 3.4 Long-term variation in PBL thermodynamic stability over NCP

On a synoptic scale, the PBL stability is influenced by the meridional wind direction to a certain extent, which has been evidenced by the two-day detailed aircraft measurements and one-month model simulations. Furthermore, the linkage between synoptic pattern (meridional wind intensity) and PBL thermal structure is analyzed from a climatological view, which is critical for the examination of aerosol vertical distribution and ARE. The potential impact of inter-annual climate
variability on the PBL thermodynamic stability is explored in this study over the NCP during the wintertime (December, January and February) of 1980-2013. In fact, the inter-annual variabilities of the PBL are affected by numerous factors, and we only discuss the influence of meridional wind intensity in this study, which is a climatological extension of the case study. Following the method developed by Wu and Wang (2002), the index of SH and the index of EAWM are used to analyze the
possible factors that influence the PBL thermal structure over the NCP. The index of SH and EAWM are the two variables describing the meridional wind intensity over the coast of East Asia, which are crucial to the PBL thermal structure. The PBL lapse rate is calculated in domain d03 depicted in Fig. 1b between 1000 hPa and 850 hPa.

Fig. 15 suggests that the relationships between SH index, EAWM index, and PBL lapse rate are highly correlated. The results indicate that the inter-annual variabilities of the EAWM and SH, which are indicators of the meridional wind
intensity, are closely related to the PBL thermodynamic stability over the NCP region. Weak EAWM and SH, in conjunction with the prevailing south wind, decrease the lapse rate between 1000 hPa and 850 hPa and stabilize the PBL, while the strong EAWM and SH play the opposite role. According to the above synoptic analysis, it is concluded that the changes in PBL properties will further affect the surface air quality, aerosol vertical distribution, and ARE. As a result, the EAWM and SH indirectly influence the aerosol vertical distribution by modulating the PBL thermodynamic structure.

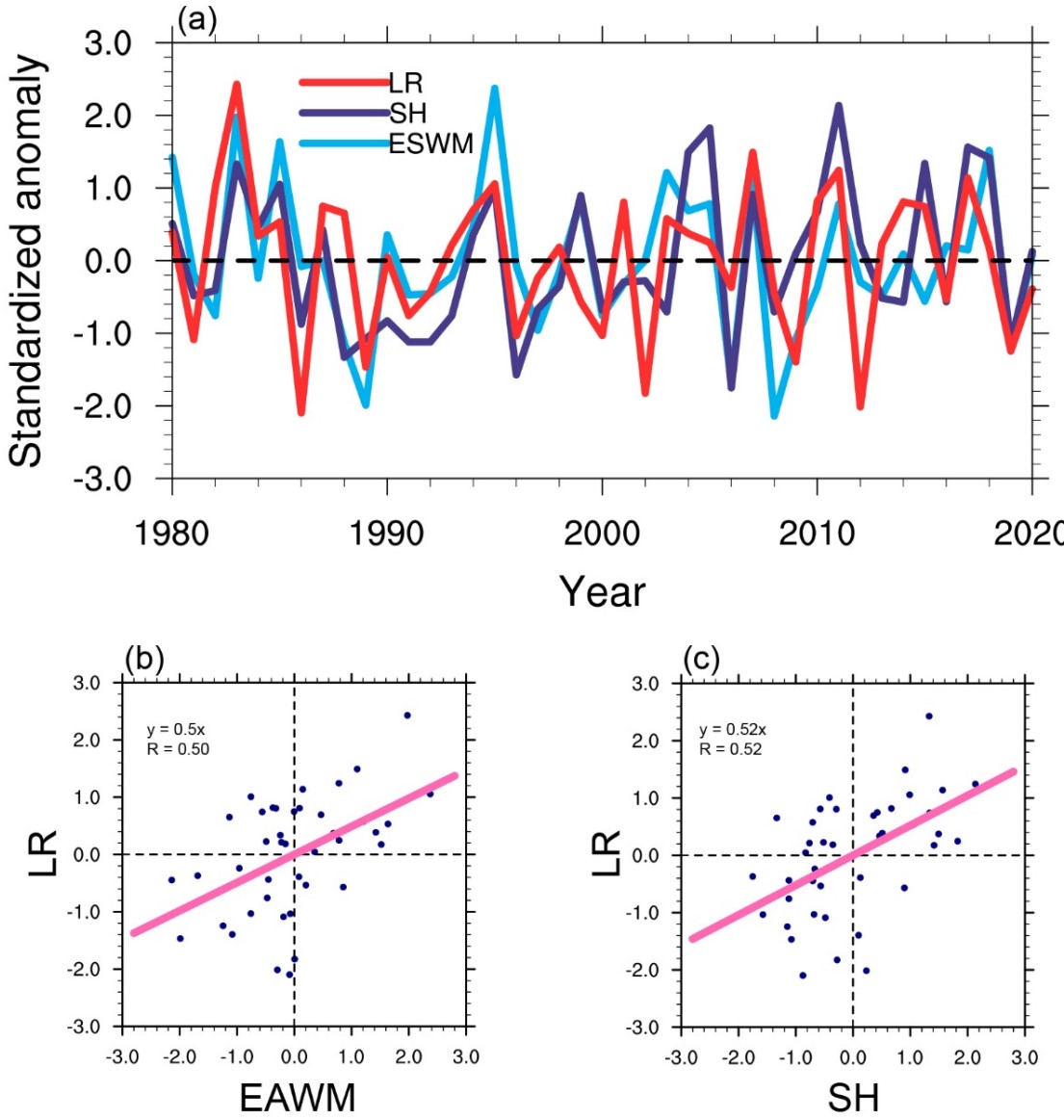

**Figure 15: (a) Time series of the standardized anomaly of the wintertime boundary layer lapse rate (LR) in domain d03 between 1000 hPa and 850 hPa, the index of Siberian High (SH), and the index of East Asian Winter Mooson (EAWM) from 1980 to 2020. The scatter plots of the correlations between (b) the standardized anomaly of the ESWM and LR, as well as between (c) the standardized anomaly of the SH and LR. Standardized anomaly is calculated by dividing anomalies by the climatological standard**

**deviation.**

**4 Conclusions and Summary**

In this study, the complex relationships among the large-scale synoptic patterns, local PBL thermal structures, aerosol vertical distributions, and AREs of different aerosol types are investigated by combining aircraft observations, surface measurements, reanalysis data, and WRF-Chem simulations. The in-situ aircraft observations were carried out over the Beijing and Baoding Cities during the daytimes on January 3 and 4, 2020, and the numerical simulations were performed over NCP from 3 to 30 January. By and large, the meteorological variables and air pollutants are well simulated when validated by the observations.

Observations show that the surface air pollution over the Baoding City on 3 January is heavier than that on 4 January, and the aerosols are constrained to the low layer on 3 January while the aerosols mix more homogeneous vertically on 4 January. The numerical simulations suggest that the synoptic pattern over the Baoding City differs between the two days, and the prevailing wind direction is opposite with a southwest wind on 3 January and a northeast wind on 4 January. Through the two-day detailed measurement and a long-term simulation, we conclude that the synoptic condition may affect the PBL thermal structure, thus affecting the vertical dispersion of aerosols. The south winds flow the warm and polluted air to the NCP and stabilize the PBL, hence exacerbating the surface air pollution. The north winds flow the cold and clean air to the NCP, establishing an unstable stratification and transporting the pollutants to the upper layer.

The suspended aerosols may also modify the PBL thermodynamic stability through ARE to some extent, and the synoptic condition can modulate the sensitivity of PBL-aerosol feedback by influencing the PBL thermal structure and aerosol vertical distribution. The sensitive numerical experiments reveal that the absorbing and scattering aerosols have different effects when modifying the PBL thermal structure, and the AREs are also influenced by the aerosol vertical distribution. Aerosol-PBL feedback of absorbing aerosols is highly dependent on its vertical distribution. In the case of stable weather patterns under the influence of the south wind, the absorbing aerosols concentrated in the low layer have a strong radiative heating effect on the atmosphere (stove effect). This stove effect disturbs the stratification stability of the low layer, promotes the development of the PBL in coordination with the surface sensible heat flux, and transports heat and pollutants to the upper layer through turbulence. In the case of unstable weather patterns under the influence of the north wind, the absorbing aerosols in the upper atmosphere heat the atmosphere (dome effect) and inhibit the development of the PBL, which further leads to the accumulation of surface pollutants. The inhibition effect of scattering aerosols on the PBL is independent of the aerosol height distribution but only depends on its concentration. Scattering aerosols reflect solar radiation back to the space (umbrella effect), thus weakening the surface sensible heat flux and upward heat transfer, inhibiting the development of the PBL and further accumulating surface pollutants. In summary, the umbrella, stove and dome effects of the scattering and absorbing aerosols under different synoptic patterns are illustrated with four scenarios in Fig. 16, which corresponds to the conclusion indicated in the four subgraphs of Fig. 13. The conclusions enlighten that we should mainly control the emissions of scattering aerosols under the stable stratification, and cooperate to control the emissions of scattering and absorbing aerosols in an unstable stratification, especially to prevent the accumulation of surface pollutants caused by the dome effect of absorbing aerosols.

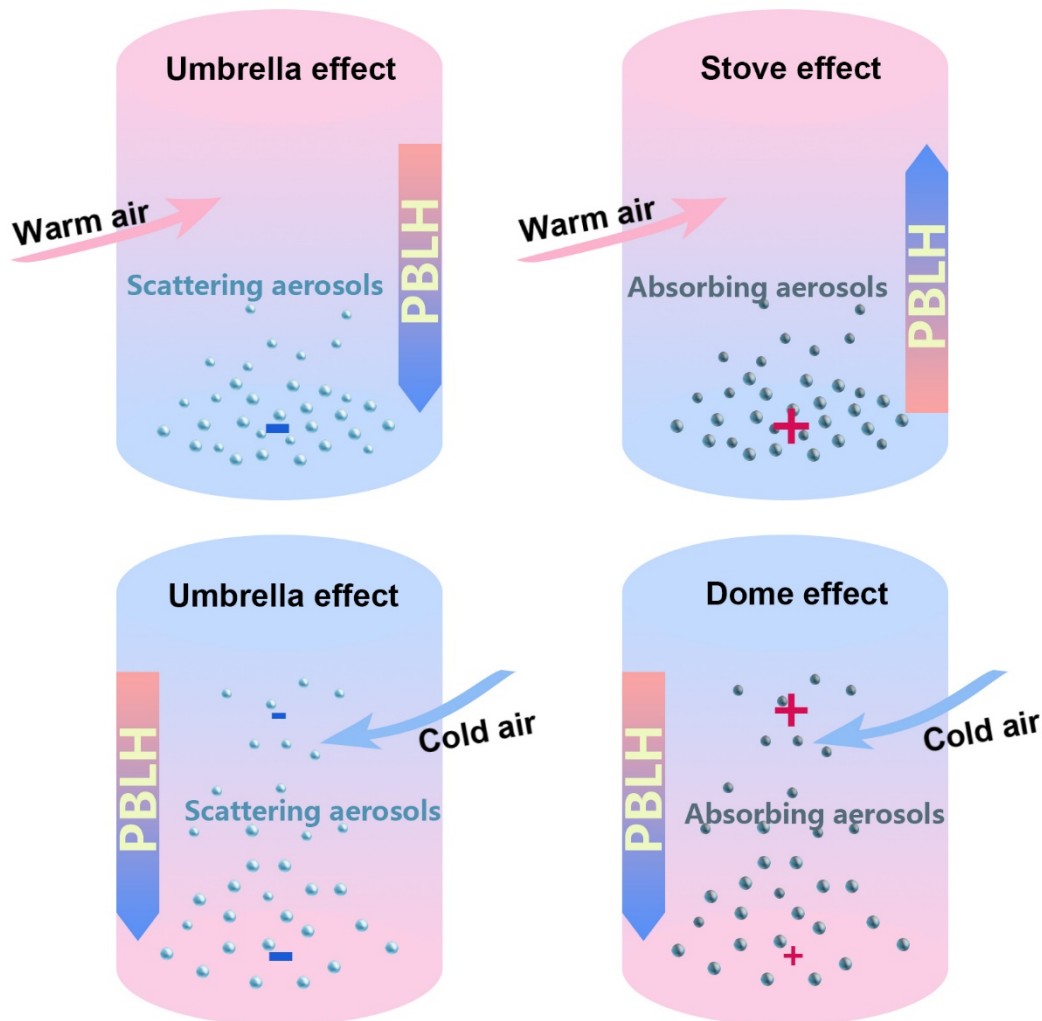

495

**Figure 16: The illustration of the umbrella, stove, and dome effects of the scattering and absorbing aerosols under different synoptic patterns. The red plus and blue minus signs indicate the aerosol heating and dimming effect, respectively. The upward and downward arrows denote the increase and decrease in PBLH caused by ARE, respectively.**

In addition, the factors affecting the wintertime PBL stability over the NCP area are further analyzed from the climatological perspective, which is mainly controlled by the intensity of EAWM and SH. A strong winter monsoon creates an unstable PBL stratification, whereas a weak winter monsoon stabilizes the PBL. Combined with the umbrella, stove, and dome effects of scattering and absorbing aerosols under different synoptic conditions, this finding aids us in determining which pollutants to target in different monsoon years and achieving more precise air pollution control.

It should be noted that only two days of pollution episodes are measured in this study. While the model simulations produce evidence that is compatible with the observations and substantiate the observational analysis, additional measurements should be carried out in the near future.

## Code/Data availability

The surface observational air pollutant data are collected from http://www.cnemc.cn. The meteorological data taken from the ERA5 reanalysis dataset are obtained from https://www.ecmwf.int/en/forecasts/datasets/reanalysis-datasets/era5. The NCEP-FNL global reanalysis dataset used in the WRF-Chem model is available from https://rda.ucar.edu/datasets/ds083.2. The MEIC anthropogenic emission inventories are available from www.meicmodel.org, and for more information, please contact Q. Zhang (qiangzhang@tsinghua.edu.cn). The WRF-Chem model version 3.9 is available from https://www2.mmm.ucar.edu/wrf/users/download/get_source.html. The aircraft data and surface meteorological data are available from the corresponding author. The codes that support the findings of this study are available from the corresponding author upon reasonable request.

## Author Contributions

Y. H. designed research and edited the paper; H. L., L. D., Y. C., Y. Z., M. H., and Y. H. performed research and analyzed data; D. Z., M. H., D. D., J. L., and T. M. participated in the discussions. H. L. wrote the manuscript.

## Competing Interests

The authors declare they have no conflict of interest.

## Acknowledgements

This work was jointly supported by the National Natural Science Foundation of China (Grant Nos. 41775026, 42027804, 41075012 and 40805006). We thank the three anonymous reviewers, for their constructive comments that have helped us to improve the paper.

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
