# Peer review of "Interaction between aerosol and thermodynamic stability within the PBL during the wintertime over the North China Plain: Aircraft observation and WRF-Chem simulation"

_Atmospheric Chemistry and Physics, 2021_

## Author Comment (AC1)

**Responses to comments of "Interaction between aerosol and thermodynamic stability within the PBL during the wintertime over the North China Plain: Aircraft observation and WRF-Chem simulation [Preprint acp-2021-769]" to** *Atmospheric Chemistry and Physics.*

Hao Luo, Yong Han*, and co-authors

We would like to thank the editor Dr. Li, Z. and the reviewers for giving constructive criticisms and comments, which are very helpful in improving the quality of the manuscript. We have made the point-by-point response to the comments below and revised the manuscript accordingly. We hope that the revised version can meet the favorable approval and journal requirements. The referee's comments are reproduced (*black, italic*) along with our replies (blue) and changes made to the text (red) in the revised manuscript. All the authors have read the revised manuscript and agreed with the submission in its revised form. Please check them.

**Responses to Reviewers**

**Anonymous Referee #3**

**General comments:**

*The manuscript presents the interaction of absorbing and scattering aerosol with PBL under different synoptic patterns based on the observation and WRF-Chem model simulations. It's an interesting study that help understand how such aerosol-PBL interactions affect the PBL thermodynamics and air quality (PM2.5). This paper also discussed potential impact of synoptic weather conditions on the PBL thermodynamics. Overall, the paper is written well. But, there are some confusions or missing parts that need be clarified.*

**Response:**

Dear Reviewer,

Many thanks for your positive comments and valuable suggestions. We have made the point-by-point response to the comments below and revised the manuscript according to your substantive comments, which helps improve the quality of this paper.

**Specific comments:**

**Comment NO.1:**

*1. The paper was lack of describing how aerosols optical parameters (absorption and scattering coefficient) are obtained and constrained in the WRF-Chem model in order to estimate aerosol-radiative-effect (ARE). For instance, how large are those aerosol absorption coefficients for black carbon (BC) and single scattering albedo (SSA) used?*

**Response:** Thank you for your valuable comments and insightful suggestions. In the revision, we have added a detailed description of the approach for obtaining and constraining aerosol optical parameters.

**Changes in Manuscript:**

**[Pages 6-7 Lines 168-173 (in the "Track Changes" version)]**

"The extinction, single-scattering albedo, and asymmetry factor of aerosols were computed as a function of wavelength and three-dimensional positions. Each chemical constituent of the aerosol was linked to complex indices of refraction. The refractive indices of the aerosols were calculated using volume averaging for each size bin, and the Mie theory was used to derive the extinction efficiency, the scattering efficiency, and the intermediate asymmetry factor. Aerosol optical properties were then determined by summarizing all size bins (Fast et al. 2006). The refractive indices of various aerosol components were reported in Barnard et al. (2010)."

**References**

Barnard, J. C., Fast, J. D., Paredes-Miranda, G., Arnott, W. P., and Laskin, A.: Technical Note: Evaluation of the WRF-Chem "Aerosol Chemical to Aerosol Optical Properties" Module using data from the MILAGRO campaign, Atmos. Chem. Phys., 10, 7325-7340, 10.5194/acp-10-7325-2010, 2010.

Fast, J. D., Gustafson Jr., W. I., Easter, R. C., Zaveri, R. A., Barnard, J. C., Chapman, E. G., Grell, G. A., and Peckham, S. E.: Evolution of ozone, particulates, and aerosol direct radiative forcing in the vicinity of Houston using a fully coupled meteorology-chemistry-aerosol model, Journal of Geophysical Research: Atmospheres, 111, https://doi.org/10.1029/2005JD006721, 2006.

**Comment NO.2:**

*2. The paper did not clearly show the PBL-height until they were given in Fig.10 in Page-19. Please describe how the PBL-height is calculated or obtained? Is it based on the Richardson number or potential temperature gradient method, or else? It might be better to display the PBLH in Fig.7 or earlier.*

**Response:** Thank you for your constructive comments. PBLH in the YSU scheme is determined from the Richardson bulk number method, which has been described in the revision. As you suggested, the PBLH has been moved forward and displayed in Fig. 4.

**Changes in Manuscript:**

**[Page 7 Lines 165-166 (in the "Track Changes" version)]**

"PBLH in the YSU scheme is determined from the Richardson bulk number method."

[Figure]

**Figure 4: Temporal evolution of the EXP_Ctrl simulated temperature profile over the Baoding City. The pink and blue dashed boxes correspond to the aircraft observation periods on 3 and 4 January, respectively. The black line indicates the planetary boundary layer height (PBLH).**

**Comment NO.3:**

*3. The lapse rate is used for many times in the paper. What range of the altitude is it calculated or referred? In Line 376-377, it is referred the range between 1000 mbar and 850-mbar, but in Line 355it is referred below 1.5 km altitude.*

**Response:** Thank you for your careful comments. Due to the difference in height coordinates of the measurement and reanalysis data, in this study, the lapse rate is calculated below 1.5 km altitude by the aircraft measurement, and the lapse rate is calculated between 1000 hPa and 850 hPa by the ERA5 reanalysis data when analyzing the long-term trend. The height between 1000 hPa and 850 hPa is approximately equivalent to the height below 1.5 km. We have illustrated it in the revision.

**Changes in Manuscript:**

**[Page 6 Lines 138-139 (in the "Track Changes" version)]**

"The height between 1000 hPa and 850 hPa is approximately equivalent to the height below 1.5 km."

**Comment NO.4:**

*4. Fig. 2 (d)-(e), why are there no temperature and wind data below 750-m from the aircraft observation which are critical to assess the PBL height and thermal stability? Fig. 2 (e), why do the observed wind speeds show so large fluctuations and many stratified structures below 3 –km?*

**Response:** Thank you for your careful comments. The aircraft observed data in Baoding city is from 650 m to 3000 m, which has been shown in Fig. 1(c)-(d). Therefore, the validations of temperature and wind are above 650 m. We regret that the aircraft observed data of temperature and wind speed were not accessible below 650 m, but the

vertical profile of temperature and wind speed below 650 m can be judged through the validated simulation profile. The large fluctuations and stratified structures may be due to the influence of the aircraft itself on wind speed measurement during the circling flight. Nevertheless, the observed wind speed profile shows a consistent trend with the simulation.

**Comment NO.5:**

*5. Fig.4, it might be better to give the horizontal wind speed, wind direction and vertical wind velocity on Jan.3 and 4 as shown for the temperature.*

**Response:** Thank you for your comment. The detailed horizontal wind speed, wind direction and vertical wind velocity have already shown in Fig. 5.

**Comment NO.6:**

*6. Fig.6 (a)-(c), what are those the horizontal dash lines? Fig.6, the observed aerosol number density profiles on Jan.3 and Jan.4 show very similar stratification structures below 1 km altitude except higher concentration on Jan.3. Was the aloft aerosol layer at 1.0-1.7 km altitude on Jan.4 related to the PBL vertical mixing transport?*

**Response:** Thank you for your constructive comments. The horizontal dash lines are the reference altitude of 650 m, which have been illustrated in the revision. In addition, we agree with your comment that the aloft aerosol layer between 1.0 km and 1.7 km altitude on 4 January is related to the PBL vertical mixing transport, which has been incorporated into the revision.

Changes in Manuscript:

**[Page 16 Lines 295-298 (in the "Track Changes" version)]**

"Figure 6: Aircraft observations of aerosol vertical distributions in the afternoon during the flight. (a) the aerosol number concentrations on 3 and 4 January, the dashed and solid lines denote the observations over Beijing and Baoding, respectively; (b-c) the aerosol size distributions on 3 and 4 January, respectively. The shaded areas in (a) indicate the error bars (standard deviation). The horizontal black dashed lines are the reference altitude of 650 m."

**[Page 16 Lines 289-290 (in the "Track Changes" version)]**

"The aloft aerosol layer between 1.0 km and 1.7 km altitude on 4 January is related to the PBL vertical mixing transport. The great difference in lapse rate shown in Fig. 4 leads to the disparity in atmospheric stability and aerosol dispersion ability."

**Comment NO.7:**

*7. Fig.8, How do the results show strong heating effects from BC and aerosol cooling effects in the night of Jan 3 and early morning of Jan 4 when there were lack of solar radiance?*

**Response:** Thank you for your insightful comment. Figure 8 shows the temperature difference caused by the aerosol radiation effect. The heat absorbed by BC during the daytime is stored in the atmosphere, making the temperature higher at night than there is no BC radiation effect. The cooling effect of scattering aerosol during the daytime reduces the longwave radiation of the surface at night, resulting in the decrease of air temperature.

**Comment NO.8:**

*8. Line 157, Which metric or parameter is used to quantify PBL thermodynamic stability?*

**Response:** Thank you for your critical comment. The lapse rate below 1.5 km was used to quantify the PBL thermodynamic stability. We have revised it.

Changes in Manuscript:

**[Page 7 Lines 175-176 (in the "Track Changes" version)]**

"The lapse rate below 1.5 km was used to quantify the PBL thermodynamic stability."

**Comment NO.9:**

*9. Line 265-275, authors pointed out that the PBL thermal stability potentially related to the simulated aerosol vertical distribution in Fig.7, but there are no potential temperature and winds displayed. Please give them for better understanding the discussions in Line 265-275 if possible.*

**Response:** Thank you for your valuable comment. The temperature and wind profiles have already shown in Fig. 4 and Fig. 5, which are the important factors for the discussion of Fig. 7 as you mentioned. The detailed information is on Pages 14-15 Lines 257-281 in the "Track Changes" version.

**Comment NO.10:**

*10. Fig.13, how to calculate the standardized anomaly of the wintertime boundary layer lapse rate?*

**Response:** Thank you for your constructive comment. Standardized anomaly is calculated by dividing anomalies by the climatological standard deviation. We have described it in the revision.

Changes in Manuscript:

**[Page 30 Lines 491-495 (in the "Track Changes" version)]**

"Figure 15: (a) Time series of the standardized anomaly of the wintertime boundary layer lapse rate (LR) in domain d03 between 1000 hPa and 850 hPa, the index of Siberian High (SH), and the index of East Asian Winter Mooson (EAWM) from 1980 to 2020. The scatter plots of the correlations between (b) the standardized anomaly of the ESWM and LR, as well as between (c) the standardized anomaly of the SH and LR. Standardized anomaly is calculated by dividing anomalies by the climatological standard deviation."

**Comment NO.11:**

*11.Line 289, how do you judge "stronger vertical mixing" on Jan.4 than those on Jan.3?*

**Response:** Thank you for your helpful comment. The stronger vertical mixing on 4 January is demonstrated from the temperature profile in Fig. 4, and it has been specifically revised.

Changes in Manuscript:

**[Page 19 Lines 334-335 (in the "Track Changes" version)]**

"On 4 January, due to the strong turbulence mixing (as demonstrated from the temperature profile in Fig. 4), aerosols are carried to the aloft layer."

**Comment NO.12:**

*12. Line 351, "….which are sensitive to the PBL thermal structure." The phrase "are sensitive to" might be replaced with "affect".*

**Response:** Thank you. We have corrected it.

Changes in Manuscript:

**[Page 24 Lines 396-397 (in the "Track Changes" version)]**

"In particular, the absorptive BC aerosols have both stove and dome effects, which affect the PBL thermal structure."

**Comment NO.13:**

*13. Line 389, please take out the word "systematically". This study only shows the analysis for the two-day data.*

**Response:** Thank you. We have corrected it.

Changes in Manuscript:

**[Page 31 Lines 497-499 (in the "Track Changes" version)]**

"In this study, the complex relationships among the large-scale synoptic patterns, local PBL thermal structures, aerosol vertical distributions, and AREs of different aerosol types are investigated by combining aircraft observations, surface measurements, reanalysis data, and WRF-Chem simulations."

**Comment NO.14:**

*14. The paper only considers the PBL thermodynamics and turbulent mixing related to the lapse rate and temperature variation, but ignores temperature-RH related secondary aerosols formation (SOA) on Jan.3 and 4 such as nitrate and sulfate SOA.*

**Response:** Thank you for your insightful comment. In this study, our primary interest is to distinguish the aerosol-PBL interaction of absorbing and scattering aerosols under contrasting synoptic patterns and aerosol vertical distributions. We mainly focus on the absorbing/scattering aerosol vertical distribution and their contrasting radiation effect.

The temperature-RH related secondary aerosol formation is a rather important issue that needs to be investigated. In the WRF-Chem simulation, the secondary aerosol formation has been included.

Again, we would like to thank you for taking your time to review this manuscript and providing insightful comments and advice. we believe that this work has been much improved with your constructive and informative remarks.

Dr. Yong Han

On behalf of all the authors

---

## Author Comment (AC3)

Responses to comments of "Interaction between aerosol and thermodynamic stability within the PBL during the wintertime over the North China Plain: Aircraft observation and WRF-Chem simulation [Preprint acp-2021-769]" to *Atmospheric Chemistry and Physics*.

Hao Luo, Yong Han\*, and co-authors

We would like to thank the editor Dr. Li, Z. and the reviewers for giving constructive criticisms and comments, which are very helpful in improving the quality of the manuscript. We have made the point-by-point response to the comments below and revised the manuscript accordingly. We hope that the revised version can meet the favorable approval and journal requirements. The referee's comments are reproduced (*black, italic*) along with our replies (blue) and changes made to the text (red) in the revised manuscript. All the authors have read the revised manuscript and agreed with the submission in its revised form. Please check them.

**Responses to Reviewers**

**Anonymous Referee #1**

The study tries to distinguish the aerosol-PBL interaction of absorbing and scattering aerosols under contrasting synoptic patterns and aerosol vertical distributions. They use aircraft measurements model simulations to estimate the aerosol radiative effects over the North China Plain. In general, this manuscript investigated an interesting topic with ample analyses. A concept scheme is further summarized to describe their findings. However, the significance and representative of this study should be carefully discussed. This manuscript must address several major issues, before the potential publication.

**Response:**

**Dear Reviewer,**

We would like to thank you for your time in reviewing this manuscript. Many thanks for your meticulous judgments and suggestions, which are very helpful in improving our manuscript. We have made the point-by-point response to the comments below and revised the manuscript according to your substantive comments, which helps improve the quality of this paper. The revision mainly includes long-term modeling for a more robust and representative conclusion.

**Major Comments:**

**Comment NO.1&2:**

1. Based on the model and aircraft data, this manuscript presents a case study for two days. It is questionable whether the conclusions from the case study are representative. The authors even draw a concept scheme from the case analyses. I believe the robustness of the conclusions needs to be carefully discussed.

2. Only two cases of aerosol vertical distribution are discussed. However, the aerosol vertical distribution varies greatly case by case. It may not be feasible to discuss the impacts of synoptic conditions on the aerosol vertical distribution. It may not be valid to draw a meaningful conclusion about aerosol stratifications and absorptions based on the two cases.

**Response:** Thank you for your critical comments and insightful suggestions. Your first two remarks are both on the robustness of the conclusions based on the two cases, so we here reply to them together. In the revision, we have added the statistical analysis of a long-term simulation in Baoding city for nearly one month from January 3 to 30, 2020. The long-term simulation results give a more robust and representative conclusion, which can compensate for the two-day case investigation. Please see the revision below for the details.

**Changes in Manuscript:**

**[Pages 24-29 Lines 394-469 (in the "Track Changes" version)]**

3.3.3 Statistical properties of the PBL and AREs under different synoptic conditions

It is noticeable that different aerosol vertical distributions between the two days contribute to distinct AREs due to the synoptic condition and PBL thermal stability differences from the measurements and simulations. In particular, the absorptive BC aerosols have both stove and dome effects, which affect the PBL thermal structure. Here, we further analyze the modeling results for nearly one month from January 3 to 30, 2020 in Baoding city to give a more significant and representative conclusion.

3

Fig. 11 shows the correlations between the daily average 10 m meridional wind speed, lapse rate within 1.5 km, and PBLH. The negative correlation between the 10 m meridional wind speed and the lapse rate within 1.5 km (Fig. 11a) suggests that the increased south wind stabilizes the PBL, whereas the strong north wind destabilizes the PBL. The variation in lapse rate has a direct impact on the development of the PBL, as evidenced by the PBLH modification shown in Fig. 11b. Fig. 12 compares the distinct vertical distributions of aerosols caused by north and south winds. Samples with a daily average wind speed within  $\pm 0.05$  m s-1 are discarded to avoid the north-south reverse of wind direction in a day. Eventually, 16 days with the prevailing north wind and 8 days with the prevailing south wind are used. The result indicates that the synoptic condition influences the PBL thermal structure, thereby affecting the vertical dispersion of aerosols. The warm and polluted air is carried to the NCP by the south winds, which stabilize the PBL, exacerbating the surface air pollution. The cold and clean air is carried to the NCP by the north winds, forming an unstable stratification and transporting pollutants to the upper layer.

Figure 11: Scatter plots of the correlations between (a) 10 m meridional wind speed (positive: south wind; negative: north wind) and lapse rate within 1.5 km and (b) lapse rate within 1.5 km and PBLH. The data are daily averages from January 3 to 30, 2020.

Figure 12: Vertical distributions of the aerosol number concentrations (particle diameter: 0.15-2.5 μm) under the prevailing south wind and north wind, respectively. The shaded areas indicate the error bars (standard deviation).

When evaluating the AREs of light-absorbing and light-scattering aerosols, the temperature profile variations show various patterns due to differences in aerosol concentration and vertical distribution caused by synoptic conditions, particularly the wind direction. Fig. 13 demonstrates that light-absorbing aerosols heat the atmosphere while light-scattering aerosols contribute to a cooling effect. Aerosols are constrained to the low layer under south wind conditions, and the BC aerosols result in a warming effect below 1 km (stove effect), while the scattering aerosols cool the layer below 0.6 km (umbrella effect). In contrast, the PBLs exhibit strong turbulence mixing when influenced by the north winds, and the aerosols are carried to the aloft layer. The aloft scattering particles prevent incident solar radiation from reaching the low layer, resulting in cooling effects below 1 km (umbrella effect), whereas the aloft absorbing aerosols heat the upper layer between 0.5 and 1.5 km (dome effect). The remarkable aerosol effects under south winds attribute to the accumulation of aerosols under adverse weather conditions. The contrasting aerosol vertical distributions caused by the varying synoptic conditions lead to different AREs, which is consistent with the results obtained on January 3 and 4, 2020.

---

## Author Response (AR1)

**Responses to comments of "Interaction between aerosol and thermodynamic stability within the PBL during the wintertime over the North China Plain: Aircraft observation and WRF-Chem simulation [Preprint acp-2021-769]" to** *Atmospheric Chemistry and Physics.*

Hao Luo, Yong Han*, and co-authors

We would like to thank the editor Dr. Li, Z. and the reviewers for giving constructive criticisms and comments, which are very helpful in improving the quality of the manuscript. We have made the point-by-point response to the comments below and revised the manuscript accordingly. We hope that the revised version can meet the favorable approval and journal requirements. The referee's comments are reproduced (*black, italic*) along with our replies (blue) and changes made to the text (red) in the revised manuscript. All the authors have read the revised manuscript and agreed with the submission in its revised form. Please check them.

**Responses to Reviewers**

**Anonymous Referee #1**

*The study tries to distinguish the aerosol-PBL interaction of absorbing and scattering aerosols under contrasting synoptic patterns and aerosol vertical distributions. They use aircraft measurements model simulations to estimate the aerosol radiative effects over the North China Plain. In general, this manuscript investigated an interesting topic with ample analyses. A concept scheme is further summarized to describe their findings. However, the significance and representative of this study should be carefully discussed. This manuscript must address several major issues, before the potential publication.*

**Response:**

Dear Reviewer,

We would like to thank you for your time in reviewing this manuscript. Many thanks for your meticulous judgments and suggestions, which are very helpful in improving our manuscript. We have made the point-by-point response to the comments below and revised the manuscript according to your substantive comments, which helps improve the quality of this paper. The revision mainly includes long-term modeling for a more robust and representative conclusion.

**Major Comments:**

**Comment NO.1&2:**

*1.      Based on the model and aircraft data, this manuscript presents a case study for two days. It is questionable whether the conclusions from the case study are*

*representative. The authors even draw a concept scheme from the case analyses. I believe the robustness of the conclusions needs to be carefully discussed.*

*2.      Only two cases of aerosol vertical distribution are discussed. However, the aerosol vertical distribution varies greatly case by case. It may not be feasible to discuss the impacts of synoptic conditions on the aerosol vertical distribution. It may not be valid to draw a meaningful conclusion about aerosol stratifications and absorptions based on the two cases.*

**Response:** Thank you for your critical comments and insightful suggestions. Your first two remarks are both on the robustness of the conclusions based on the two cases, so we here reply to them together. In the revision, we have added the statistical analysis of a long-term simulation in Baoding city for nearly one month from January 3 to 30, 2020. The long-term simulation results give a more robust and representative conclusion, which can compensate for the two-day case investigation. Please see the revision below for the details.

**Changes in Manuscript:**

**[Pages 24-29 Lines 394-469 (in the "Track Changes" version)]**

[revised manuscript text omitted]

**Comment NO.3:**

*3.      There are large diurnal variations in PBL. However, figure 6 only mentions the date. How about the specific time? As aerosol vertical distribution in figure 6 may not represent the daily condition, the authors need to address the diurnal changes in PBL and aerosol vertical distribution.*

**Response:** Thank you for your valuable comments. The specific time has been added in the revision, which is in the afternoon during the flight on 3 and 4 January. Table 1 also gives the specific take-off and landing time. Due to the aircraft measurement cannot characterize the diurnal variations in PBL and aerosol vertical distribution, we address the diurnal variations based on simulation results, which are shown in Fig. 7 and Fig. 10.

**Changes in Manuscript:**

**[Page 15 Lines 283-284 (in the "Track Changes" version)]**

"As demonstrated in Fig. 6, contrasting aerosol vertical distributions are observed with aircraft during the afternoon of January 3 and 4, 2020, and the specific times are shown in Table 1."

**Comment NO.4:**

*4.     After the case study, the manuscript presents the long-term variation in PBL thermodynamic stability. However, I feel this part is disconnected from the main analyses. The long-term changes in PBL are affected by numerous factors. I did not find any useful conclusion from the analyses. The analyses also cannot conclude that "the inter-annual variability of the EAWM and SH can influence aerosol vertical distribution and ARE", which is cited from Section 3.4.*

**Response:** Thank you for your critical comments. We agree with you that the PBL is affected by numerous factors, but we mainly address the relation between PBL and meridional wind intensity in this study. Despite the two-day case study, substantial one-month simulation results in the revision indicate that the PBL stability is closely related to the synoptic pattern, especially the meridional wind direction. These results are attained from the synoptic-scale condition. Furthermore, section 3.4 aims to determine whether the same conclusion can be reached from a climatological standpoint. EAWM and SH are the two indicators of meridional wind intensity over the NCP, so it is valuable to analyze the relations between EAWM, SH and PBL stability from long-term datasets. In the revision, we have added the linkage between the discussions of long-term variations in PBL and the results of two-day measurements and one-month simulations, due to the contextual disconnects in the previous manuscript. In addition,

the sentence "the inter-annual variability of the EAWM and SH can influence the aerosol vertical distribution and ARE" has been rewritten.

**Changes in Manuscript:**

**[Page 29 Lines 471-474 (in the "Track Changes" version)]**

"On a synoptic scale, the PBL stability is influenced by the meridional wind direction to a certain extent, which has been evidenced by the two-day detailed aircraft measurements and one-month model simulations. Furthermore, the linkage between synoptic pattern (meridional wind intensity) and PBL thermal structure is analyzed from a climatological view, which is critical for the examination of aerosol vertical distribution and ARE."

**[Pages 29-30 Lines 483-488 (in the "Track Changes" version)]**

"The results indicate that the inter-annual variabilities of the EAWM and SH, which are indicators of the meridional wind intensity, are closely related to the PBL thermodynamic stability over the NCP region. According to the previous synoptic analysis, it is concluded that the changes in PBL properties will further affect the surface air quality, aerosol vertical distribution, and ARE."

**Comment NO.5:**

5.      *Figure 14 seems to describe common sense. It is well-known that synoptic patterns, PBL thermodynamics, and aerosol vertical distribution can affect each other. What is the significance of Figure 14?*

**Response:** Thank you for your constructive criticisms. As you suggested, Fig. 14 has been removed from the revised version since it cannot contribute any further information.

**Comment NO.6:**

*6.    Figure 15 tries to summarize the impacts of aerosol on PBL under different synoptic patterns. However, the "cold/warm advection" is only one factor and cannot fully represent the synoptic conditions. Four different scenarios are discussed but do not well support by their analyses.*

**Response:** Thank you for your valuable comments. We agree with your concerns about the disconnection between the concept scheme and the analyses. In the revision, a long-term simulation result gives a robust conclusion shown in Fig. 13, which deepens the connection between the concept scheme and the analyses. Fig. 13 demonstrates that light-absorbing and light-scattering aerosols contribute to distinct AREs under different synoptic patterns, where the synoptic pattern is categorized by the cold/warm advection. Therefore, the "cold/warm advection" represents the synoptic condition in the concept scheme, and we agree with you that the advection is not the only factor. Four different scenarios summarized in Fig.16 (Fig. 15 in the original version) can well correspond to the four subgraphs in Fig. 13.

**Changes in Manuscript:**

**[Page 32 Lines 528-530 (in the "Track Changes" version)]**

"In summary, the umbrella, stove and dome effects of the scattering and absorbing aerosols under different synoptic patterns are illustrated with four scenarios in Fig. 16, which corresponds to the conclusion indicated in the four subgraphs of Fig. 13."

[Figure]

**Figure 13: Temporal evolution of the temperature profile variation influenced by aerosol radiative effect (ARE). (a) ARE by other aerosols (EXP_WFexBC – EXP_WoF) and (b) ARE by BC (EXP_Ctrl – EXP_WFexBC) under the prevailing south wind; (c) ARE by other aerosols and (d) ARE by BC under the prevailing North wind.**

[Figure]

**Figure 16: The illustration of the umbrella, stove, and dome effects of the scattering and absorbing aerosols under different synoptic patterns. The red plus and blue minus signs indicate the aerosol heating and dimming effect, respectively. The upward and downward arrows denote the increase and decrease in PBLH caused by ARE, respectively.**

Again, we would like to thank you for taking your time to review this manuscript and providing insightful comments and advice. we believe that this work has been much improved with your constructive and informative remarks.

Dr. Yong Han

On behalf of all the authors

**Anonymous Referee #2**

**General comments:**

*This work investigates the roles of the synoptic pattern, PBLH, aerosol type and vertical distribution in aerosol-PBL interactions by using aircraft measurements, model simulation. Several parallel numerical experiments are conducted to investigate the radiative effects of scattering and absorbing aerosols under different aerosol vertical distributions. Moreover, the long-term variation in PBL stability from 1980 to 2020 over the NCP region is examined. However, the current method and model settings in this work cannot well support the conclusion proposed, and need to be reconsidered. In addition, I personally think that hardly the case study for 2 days with a flawed method can be beneficial in determining which pollutants to target and achieving precise controls of air pollution. Here list some major concerns that need to be addressed.*

**Response:**

Dear Reviewer,

We would like to thank you for your time in reviewing this manuscript. Many thanks for your meticulous judgments and suggestions, which are very helpful in improving our manuscript. We have made the point-by-point response to the comments below and revised the manuscript according to your substantive comments, which helps improve the quality of this paper. The revision mainly includes a more extensive description of the model setup as well as a long-term modeling for a more robust conclusion.

**Major comment:**

**Comment NO.1:**

*The present study focuses on the case on 3-4 Jan 2020. However, the WRF-Chem model simulation was started on 2 Jan with only 16 hours as model spin-up. As well acknowledged, the atmospheric lifetime of aerosol is more than one week. That is to say, such a short spin-up time cannot reflect the aerosol background, chemical environment (OH radical, VOC levels and etc) and regional transport at all. Thus, it is not possible that the secondary scattering aerosol like sulfate and nitrate aerosol was well reproduced. I suggest that the authors either prolong the model time or use the other model output as the chemical initial condition.*

**Response:** Thank you for your valuable comments. We agree with your concerns about the long period (more than one week) required for secondary aerosol production. As a matter of fact, the chemical outputs from the previous modeling periods between 1200UTC 25 December 2019 and 0000UTC 2 January 2020 (7.5 days) were used as the initial chemical conditions for the modeling. We are sorry for the ambiguous statement regarding the 16-h spin-up time, which was used to achieve a quasi-steady state of the model's meteorological process. The previous modeling results of 7.5 days, which was regarded as the spin-up time for the chemistry, were discarded in the analysis. In the revision, we have added a detailed description of the spin-up time and the initial chemical condition.

**Changes in Manuscript:**

**[Page 6 Lines 149-155 (in the "Track Changes" version)]**

"The simulations were conducted from 1200UTC on December 25, 2019 to 0000UTC on January 31, 2020, with the first 7.5 days as spin-up time for chemistry. The model was run with an 84 hours model cycle, with the first 12 hours discarded as spin-up time and the last 72-hour results used for the final analysis. The chemical outputs from previous runs were used as the initial conditions for the subsequent overlapping 84-hour simulation. The simulations of the case study were carried out from 0000 UTC on January 2 to 1800 UTC on January 4, 2020 with the first 16 hours as the model spin-up time, and the chemical outputs from the previous run were used as the initial conditions."

**Comment NO.2:**

*Another issue concerning the model simulation is that the model adopted a 3-km grid resolution but used an emission inventory with ~30km grid, which is not very matched with each other in spatial. Please clarify. Besides, since that NCP has experienced significant emission reduction in past years, please specify the base year of the emission inventory that was used in this work.*

**Response:** Thank you for your critical comments. In this study, the three nesting domains with horizontal resolutions of 27, 9 and 3 km were performed. Nesting is a useful technique that can be used in WRF-Chem where a single, or several higher resolution model domains (nests) are located within a coarser, parent domain. This technique makes it possible to downscale from data with large grid space to the high-resolution scales, using the parent domain as a provider of lateral boundary conditions for the nest. The resolution of the outermost domain is 27 km, which is of the same order of magnitude as the emission inventory with a ~30km grid. Moreover, previous

studies also used a similar method in the WRF-Chem modeling with the MEIC emission inventory, which is partially listed in Table Comment 2.1.

**Table Comment 2.1. List of the domain information of the references.**

| Reference | Emission inventory | Domain |
|---|---|---|
| Sha, T., et al, STOTEN, 2021 | MEIC, 2016 | D01: 27 km; D02: 9 km |
| Shu, Z., et al, ACP, 2021 | MEIC, 2012 | D01: 48 km; D02: 12 km; D03: 3 km |
| Liu, C., et al, AE, 2021 | MEIC, 2010 | D01: 36 km; D02: 12 km; D03: 4 km |
| Qu, Y., et al, AE, 2020 | MEIC, 2010 | D01: 81 km; D02: 27 km; D03: 9 km; D04: 3 km |

In addition, as you suggested, we have included the base year (2017) of the emission inventory that was used in this study in the revised version. The MEIC in 2017 is the newest and the most accurate version that can well represent the emissions in recent years, though the NCP has experienced emission reduction in the past several years.


"The statistical validations of BC concentration vertical profiles show an *R* of 0.67 and a total MB of -0.18 μg m$^{-3}$. The statistical validations of PM$_{2.5}$ mass concentration indicate an *R* of 0.79, a total MB of -4.91 μg m$^{-3}$, and an NMB of 4.69% during the daytime (Table 3). Therefore, in this study, we consider that the WRF-Chem simulation is in line with the observation and can capture the weather characteristics as well as the general distributions and variations in air pollutants."

[Figure]

Figure 3: Validation of aerosol concentration between the modeling (EXP_Ctrl) and in-situ observations. (a) aircraft measured BC concentration vertical distributions; (b) ground-based observed PM$_{2.5}$ concentration. The shaded areas indicate the error bars (standard deviation).

**Comment NO.5:**

*As pointed out, the combination of aircraft and model simulation was not so much. I do not think that the simulation needs to confine to these two days. Hardly the case study for just two days can represent the general conditions in this small region.*

**Response:** Thank you for your valuable comments and insightful suggestions. As you suggested, in the revision, we have added the statistical analysis of a long-term simulation in Baoding city for nearly one month from January 3 to 30, 2020. The long-term simulation results give a more robust and representative conclusion, which can compensate for the two-day case investigation. Please see the revision below for the details.

**Changes in Manuscript:**

**[Pages 24-29 Lines 394-469 (in the "Track Changes" version)]**

[revised manuscript text omitted]

**Comment NO.6:**

*Figure 12 is not a good way to show the impact of PBL and pollution. PBLH cannot well reflect the structure itself. And for CO, it is a relatively long-lived species in the atmosphere with a background concentration of around 100 ppb. The short-term perturbations of aerosol on PBL just for two days cannot substantially influence the concentration since the background concentration in the lower troposphere is way larger than the perturbation caused by ARE.*

**Response:** Thank you for your constructive criticisms. The original reason we chose CO to study the effects of PBL on aerosol is that CO is a rather stable component in the atmosphere which is little impacted by other particles. We agree with your assessment that the CO concentration cannot be substantially influenced by the perturbation caused by ARE, as a result, the correlations shown in Fig. 12 are not immediately apparent. The primary aims of Figs. 11-12 were to provide a spatial statistical conclusion of the aerosol-PBL feedback, but the revision gives a long-term simulation finding that is

more representative than the prior analysis. In any case, we have replaced the prior Figs. 11-12 with the updated results presented in the response of Comment NO.5.

**Minor issues:**

**Comment NO.7:**

*Table2: NMB makes no sense when evaluating air temperature.*

**Response:** Thank you. The temperature has an MB of -0.87℃, which is within acceptable limits when compared with the previous study (Ding, A., et al., GRL, 2016, temperature MB: -1.25~1.02℃). Due to the low temperature value, the NMB is relatively higher. The correction coefficient between the ground-based (aircraft) observation and EXP_Ctrl is 0.87 (0.98) that indicating the model well describes the temperature variation.


"Figure 15: (a) Time series of the standardized anomaly of the wintertime boundary layer lapse rate (LR) in domain d03 between 1000 hPa and 850 hPa, the index of Siberian High (SH), and the index of East Asian Winter Mooson (EAWM) from 1980 to 2020. The scatter plots of the correlations between (b) the standardized anomaly of the ESWM and LR, as well as between (c) the standardized anomaly of the SH and LR. Standardized anomaly is calculated by dividing anomalies by the climatological standard deviation."

Again, we would like to thank you for taking your time to review this manuscript and providing insightful comments and advice. we believe that this work has been much improved with your constructive and informative remarks.

Dr. Yong Han
On behalf of all the authors

**Anonymous Referee #3**

**General comments:**

*The manuscript presents the interaction of absorbing and scattering aerosol with PBL under different synoptic patterns based on the observation and WRF-Chem model simulations. It's an interesting study that help understand how such aerosol-PBL interactions affect the PBL thermodynamics and air quality (PM2.5). This paper also discussed potential impact of synoptic weather conditions on the PBL thermodynamics. Overall, the paper is written well. But, there are some confusions or missing parts that need be clarified.*

**Response:**

Dear Reviewer,

Many thanks for your positive comments and valuable suggestions. We have made the point-by-point response to the comments below and revised the manuscript according to your substantive comments, which helps improve the quality of this paper.

**Specific comments:**

**Comment NO.1:**

*1. The paper was lack of describing how aerosols optical parameters (absorption and scattering coefficient) are obtained and constrained in the WRF-Chem model in order to estimate aerosol-radiative-effect (ARE). For instance, how large are those aerosol absorption coefficients for black carbon (BC) and single scattering albedo (SSA) used?*

**Response:** Thank you for your valuable comments and insightful suggestions. In the revision, we have added a detailed description of the approach for obtaining and constraining aerosol optical parameters.

**Changes in Manuscript:**

**[Pages 6-7 Lines 168-173 (in the "Track Changes" version)]**

"The extinction, single-scattering albedo, and asymmetry factor of aerosols were computed as a function of wavelength and three-dimensional positions. Each chemical constituent of the aerosol was linked to complex indices of refraction. The refractive indices of the aerosols were calculated using volume averaging for each size bin, and the Mie theory was used to derive the extinction efficiency, the scattering efficiency, and the intermediate asymmetry factor. Aerosol optical properties were then determined by summarizing all size bins (Fast et al. 2006). The refractive indices of various aerosol components were reported in Barnard et al. (2010)."

**Response:** Thank you for your insightful comment. In this study, our primary interest is to distinguish the aerosol-PBL interaction of absorbing and scattering aerosols under contrasting synoptic patterns and aerosol vertical distributions. We mainly focus on the absorbing/scattering aerosol vertical distribution and their contrasting radiation effect.

The temperature-RH related secondary aerosol formation is a rather important issue that needs to be investigated. In the WRF-Chem simulation, the secondary aerosol formation has been included.

Again, we would like to thank you for taking your time to review this manuscript and providing insightful comments and advice. we believe that this work has been much improved with your constructive and informative remarks.

Dr. Yong Han

On behalf of all the authors

---

## Author Response (AR2)

**Responses to comments of "Interaction between aerosol and thermodynamic stability within the PBL during the wintertime over the North China Plain: Aircraft observation and WRF-Chem simulation [Preprint acp-2021-769]" to** *Atmospheric Chemistry and Physics.*

Hao Luo, Yong Han*, and co-authors

We would like to thank the editor Dr. Li, Z. and the reviewers for giving constructive criticisms and comments, which are very helpful in improving the quality of the manuscript. We have made the point-by-point response to the comments (Report #3) below and revised the manuscript accordingly. We hope that the revised version can meet the favorable approval and journal requirements. The referee's comments are reproduced (*black, italic*) along with our replies (blue) and changes made to the text (red) in the revised manuscript. All the authors have read the revised manuscript and agreed with the submission in its revised form. Please check them.

**Responses to Referee**

**Anonymous Referee #1   (Report #3)**

*The authors make substantial changes to their manuscript. However, a few problems have been well addressed in the response.*

**Response:**

Dear Reviewer,

We would like to thank you for your time in reviewing this manuscript. The manuscript has been revised according to your substantive comments. Please check the point-by-point response to your remarks below.

**Comment NO.1**

*1. As discussed in Ma et al. (2020), the stove/dome effects are mainly related to the aerosol vertical distributions, which are affected by multiple factors. Horizontal advection may not lead to a certain type of aerosol vertical distribution.*

**Response:** Thank you for your critical comments and insightful suggestions. We agree with your assessment that the horizontal advection may not lead to a certain type of aerosol vertical distribution, as other factors (e.g. topography, surface cover type, mountain breeze, emission source, etc.) may also contribute to the variations in aerosol vertical dispersion (Su et al., 2018, Ma et al., 2020). Su et al. (2018) has found a nonlinearly negative response of surface particulate matter to planetary boundary layer height (PBLH) evolution over the polluted regions in China, especially in the North China Plain (NCP), which supports your opinion. However, in this study, our principal result is that the synoptic forcing (e.g. horizontal advection) may contribute to the PBL

thermal structure and PBLH, and then the PBL influences the vertical dispersion of aerosols. The previous study has also verified the conclusion that wind field and PBLH tend to be correlated, which means meteorologically favorable horizontal and vertical dispersion conditions are likely to occur together (Su et al., 2018). To make the paper clearer, some explanations have been added in the revised version.

**Reference**

Ma, Y., Ye, J., Xin, J., Zhang, W., Vilà-Guerau de Arellano, J., Wang, S., Zhao, D., Dai, L., Ma, Y., Wu, X., Xia, X., Tang, G., Wang, Y., Shen, P., Lei, Y., and Martin, S. T.: The Stove, Dome, and Umbrella Effects of Atmospheric Aerosol on the Development of the Planetary Boundary Layer in Hazy Regions, Geophysical Research Letters, 47, e2020GL087373, 10.1029/2020GL087373, 2020.

Su, T., Li, Z., and Kahn, R.: Relationships between the planetary boundary layer height and surface pollutants derived from lidar observations over China: regional pattern and influencing factors, Atmos. Chem. Phys., 18, 15921-15935, 10.5194/acp-18-15921-2018, 2018.

**Changes in Manuscript:**

**[Page 21 Lines 394-395 (in the "Track Changes" version)]**

"The previous study has also verified the conclusion that wind field and PBLH tend to be correlated, which means meteorologically favorable horizontal and vertical dispersion conditions are likely to occur together (Su et al., 2018)."

**Added Reference**

Su, T., Li, Z., and Kahn, R.: Relationships between the planetary boundary layer height and surface pollutants derived from lidar observations over China: regional pattern and influencing factors, Atmos. Chem. Phys., 18, 15921-15935, 10.5194/acp-18-15921-2018, 2018.

**Comment NO.2**

*2. Based on the schematic diagram (Figure 16), absorbing aerosols and warm advection will lead to the stove effect, which increases the PBLH. As aerosol will significantly suppress the sensible heat, it may need more elaborations. Lape rate is only one of the impact factors.*

**Response:** Thank you for your constructive criticisms. We agree with you that the stove effect of the near-surface absorbing aerosol traps the solar radiation and leads to a decrease in the surface sensible heat. Ma et al. (2020) has confirmed that the radiative heating of high concentration absorbing aerosols near the surface will offset and exceed the reduction of surface sensible heat flux, resulting in near-surface heating. The low-layer heating, together with the surface buoyancy flux, aids in the growth of PBLH (Ma et al., 2020). We have elaborated on it in the revision.

**Reference**

Ma, Y., Ye, J., Xin, J., Zhang, W., Vilà-Guerau de Arellano, J., Wang, S., Zhao, D., Dai, L., Ma, Y., Wu, X., Xia, X., Tang, G., Wang, Y., Shen, P., Lei, Y., and Martin, S. T.: The Stove, Dome, and Umbrella Effects of Atmospheric Aerosol on the Development of the Planetary Boundary Layer in Hazy Regions, Geophysical Research Letters, 47, e2020GL087373, 10.1029/2020GL087373, 2020.

**Changes in Manuscript:**

**[Page 19 Lines 356-359 (in the "Track Changes" version)]**

"Ma et al. (2020) has confirmed that the radiative heating of high concentration absorbing aerosols near the surface will offset and exceed the reduction of surface sensible heat flux, resulting in near-surface heating (stove effect). The low-layer heating,

together with the surface buoyancy flux, invigorates the turbulent mixing and leads to a higher lapse rate and PBLH (Ma et al., 2020)."

**Reference**

Ma, Y., Ye, J., Xin, J., Zhang, W., Vilà-Guerau de Arellano, J., Wang, S., Zhao, D., Dai, L., Ma, Y., Wu, X., Xia, X., Tang, G., Wang, Y., Shen, P., Lei, Y., and Martin, S. T.: The Stove, Dome, and Umbrella Effects of Atmospheric Aerosol on the Development of the Planetary Boundary Layer in Hazy Regions, Geophysical Research Letters, 47, e2020GL087373, 10.1029/2020GL087373, 2020.

**Comment NO.3**

*3. The authors may further revise the description in section 3.4. The interannual variabilities are affected by numerous factors. How the correlations conclude that "EAWM and SH with the prevailing south wind stabilize the PBL and constrain the aerosols to the near-surface, while the strong EAWM and SH play the opposite role."*

**Response:** Thank you for your valuable comments. We agree with your suggestions and have added the relevant descriptions in the revised version.

**Changes in Manuscript:**

**[Page 24 Lines 443-444 (in the "Track Changes" version)]**

"In fact, the inter-annual variabilities of the PBL are affected by numerous factors, and we only discuss the influence of meridional wind intensity in this study, which is a climatological extension of the case study."

**[Pages 24-25 Lines 453-457 (in the "Track Changes" version)]**

"Weak EAWM and SH, in conjunction with the prevailing south wind, decrease the lapse rate between 1000 hPa and 850 hPa and stabilize the PBL, while the strong

EAWM and SH play the opposite role. According to the above synoptic analysis, it is concluded that the changes in PBL properties will further affect the surface air quality, aerosol vertical distribution, and ARE. As a result, the EAWM and SH indirectly influence the aerosol vertical distribution by modulating the PBL thermodynamic structure."

Again, we would like to thank you for taking time to review this manuscript and providing insightful comments and advice. we believe that this work has been much improved with your constructive and informative remarks.

Dr. Yong Han

On behalf of all the authors